# LMAct: A Benchmark for In-Context Imitation Learning with Long Multimodal Demonstrations

Anian Ruoss [1]  Fabio Pardo [1]  Harris Chan [1]  Bonnie Li [1]  Volodymyr Mnih [1]  Tim Genewein [1]

## Abstract

In this paper, we present a benchmark to pressure-test today's frontier models' multimodal decision-making capabilities in the very long-context regime (up to one million tokens) and investigate whether these models can learn from large numbers of expert demonstrations in their context. We evaluate the performance of Claude 3.5 Sonnet, Gemini 1.5 Flash, Gemini 1.5 Pro, Gemini 2.0 Flash Experimental, GPT-4o, o1-mini, o1-preview, and o1 as policies across a battery of simple interactive decision-making tasks: playing tic-tac-toe, chess, and Atari, navigating grid worlds, solving crosswords, and controlling a simulated cheetah. We study increasing amounts of expert demonstrations in the context — from no demonstrations to 512 full episodes. Across our tasks, models rarely manage to fully reach expert performance, and often, presenting more demonstrations has little effect. Some models steadily improve with more demonstrations on a few tasks. We investigate the effect of encoding observations as text or images and the impact of chain-of-thought prompting. To help quantify the impact of other approaches and future innovations, we open source our benchmark that covers the zero-, few-, and many-shot regimes in a unified evaluation.

## 1. Introduction

The simple recipe of minimizing next-token prediction error at scale has led to large multimodal foundation models (LMs) with remarkably general capabilities (OpenAI, 2023; Anil et al., 2023; Anthropic, 2024a). Importantly, these capabilities come in two flavors: (i) the ability to produce outputs of high quality from a short and often underspecified prompt (e.g., writing an essay about a novel topic), and (ii)

the ability to learn new patterns and imitate algorithms *in context* (Reid et al., 2024; Mirchandani et al., 2023). Both types of capabilities demonstrate that LMs can manipulate learned knowledge and respond to new information in flexible and non-trivial ways. These capabilities, both necessary for reasoning and decision-making, have led to the recent surge of using LMs as agents by sampling an action from the model (Mirchandani et al., 2023; Palo & Johns, 2024).

While LMs have been shown to be able to perform non-trivial reasoning and decision-making in some domains (Huang et al., 2022; Yao et al., 2023; Romera-Paredes et al., 2024), there are also many negative results where LMs fail to perform decision-making tasks that are very simple for humans, even when LMs arguably possess great factual knowledge of the task. For example, LMs struggle to play legal moves, let alone beat amateur humans, in chess (Carlini, 2023). At the same time, state-of-the-art LMs have detailed expert knowledge of chess when queried in natural language. But this *declarative* knowledge fails to translate into effective decision-making, i.e., "know-how" (Ryle, 1949). This "knowing-doing gap" is also observed in the recently released BALROG benchmark (Paglieri et al., 2025), which evaluates zero-shot capabilities (i.e., without expert demonstrations) of state-of-the-art LMs on interactive decision-making tasks in the long time horizon setting: 5 game environments from Baby AI (Chevalier-Boisvert et al., 2019) to Nethack (Küttler et al., 2020). Overall, Paglieri et al. (2025) find that "models struggle significantly with more challenging tasks" and highlight few- and many-shot evaluation as an important potential solution. Our benchmark (developed concurrently) addresses exactly this gap by covering the full range from zero-shot to many-shot evaluation.

The general question arising from these results is whether LMs in principle have the capabilities to solve interactive decision-making tasks but misunderstand the "out-of-distribution" specification given by the zero-shot prompt or a few examples, or whether the problem is more deeply rooted. Accordingly, our paper focuses on whether conditioning on a *large* number of expert demonstrations (state-action trajectories) helps unlock the decision-making capability of pretrained LMs. In doing so, we test the multimodal in-context learning capabilities of modern LMs at their limits,

---

[1] Google DeepMind. Correspondence to: Anian Ruoss <anianr@google.com>, Tim Genewein <timgen@google.com>.

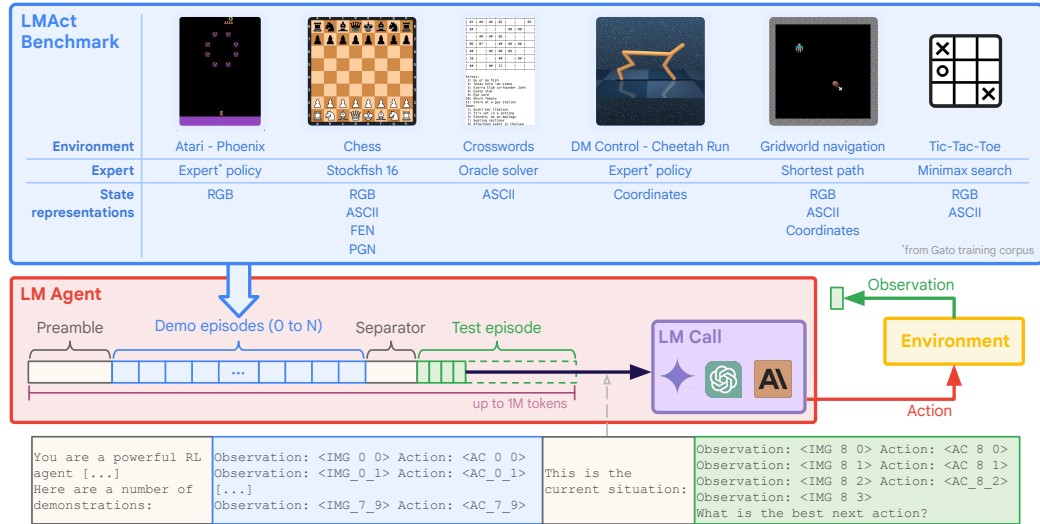

*Figure 1.* LMAct overview. Our multimodal benchmark consists of six decision-making tasks that come with an expert policy and potentially multiple state representations. For evaluation, LM performance is measured on test episodes with unseen initial states. LMs are conditioned on a generic decision-making preamble (fixed across all tasks), followed by 0 to $N$ demonstration episodes, and a separator that indicates the start of the current episode ($N$ can be up to 512, with up to 100 steps per episode, both depending on the task. The maximum context length is 1M tokens). In each step of the test episode an action is generated by the LM's predicted continuation of the context. The resulting environment interaction produces the next observation that is added to the growing context of state action pairs.

with contexts that are up to a million tokens long (with thousands of output tokens). Our main research question is:

> Can state-of-the-art LMs learn to act in dynamic environments by generalizing from *large* numbers of *multimodal* in-context expert demonstrations?

**Main Contributions** See Figure 1 for our tasks and methodology. We make the following key contributions:

- We conduct a comprehensive empirical evaluation of the multimodal in-context imitation learning capabilities of state-of-the-art LMs (Claude 3.5 Sonnet, Gemini 1.5 Flash, Gemini 1.5 Pro, Gemini 2.0 Flash Experimental, GPT-4o, o1-mini, o1-preview, and o1) on a battery of interactive, potentially long-horizon, tasks: playing tic-tac-toe, chess, and Atari, navigating grid worlds, solving crosswords, and a DM Control task.

- We show that — even when optimizing the prompt (number of demonstrations, chain-of-thought prompting, etc.) for each model and task — frontier LMs fail to reach expert performance on Atari, chess, and DM Control. Some models approach expert performance on crosswords, grid world, and tic-tac-toe. All models beat the random action baselines except on Atari.

- We vary the number of expert demonstration episodes in the context from 0 up to 512 (the limit depends on the model and the task) and find that performance is

mostly independent of the number of demonstrations. In some cases we observe strong in-context learning.

- We run a control experiment where LLM agents need to replay the single demonstration episode in the context, where all models except for o1-mini perform well.

- We open-source our in-context imitation learning benchmark that covers the zero-, few-, and many-shot regime in a unified manner, including all expert demonstrations and evaluation code at `https://github.com/google-deepmind/lm_act`.

## 2. Methods

We now briefly describe the models we evaluate (Section 2.1), our benchmark environments (Section 2.2), how we construct the prompt (Section 2.3), and our evaluation protocol (Section 2.4). Full details are given in Appendix B.

### 2.1. Models

We evaluate the current (closed-weights) frontier LMs: Claude 3.5 Sonnet (the 2024-10-22 version) (Anthropic, 2024b;c), Gemini 1.5 Flash and Gemini 1.5 Pro (the 002 versions) (Reid et al., 2024), Gemini 2.0 Flash Experimental (Google DeepMind, 2024b), GPT-4o (the 2024-08-06 version) (OpenAI, 2024a; OpenAI et al., 2024a), o1-mini and o1-preview (OpenAI, 2024c), and o1 (OpenAI et al., 2024b). Except for o1-mini and o1-preview, all models process multimodal prompts, though their exact specifica-

tions differ (see Appendix B.1). We use temperature 0 for all models (except for o1-mini, o1-preview, and o1, which have a fixed temperature of 1 (OpenAI, 2024d)). We set the maximum (output) sample length to 2048 tokens for all models (except for o1-mini, o1-preview, and o1), which is more than sufficient to achieve strong performance (see our ablation in Fig. A7). In contrast, the performance of o1-mini, o1-preview, and o1 crucially depends on the number of "reasoning tokens" (see Fig. A7), so we use a maximum (output) sample length of 8192 tokens as a good compromise between cost and performance (2048 tokens would lead to severe performance degradation on our tasks – see our ablation in Appendix C.1). We post-process the model outputs by removing all the leading/trailing white spaces and only consider the text after the keyword "Action:", i.e., we discard all (chain-of-thought) reasoning traces.

## 2.2. Environments

We consider a battery of well-known interactive decision-making environments: the Phoenix game from Atari 2600 (Bellemare et al., 2013), chess, crosswords, the cheetah run task from the DM Control suite (Tassa et al., 2018), grid world navigation, and tic-tac-toe. We briefly describe each environment below (full details in Appendix B.2). Since sampling is deterministic for most models (i.e., the temperature is 0, see Section 2.1), we introduce variability in the demonstration and evaluation episodes by varying the initial conditions (e.g., via the environment seed for DM Control or by using different openings for chess; see below).

**Atari – Phoenix**   We chose Phoenix as a representative Atari task that has somewhat dense rewards, which is important since we can only evaluate 400 frames (with action repeat 4, i.e., 100 steps), or roughly 6 seconds of play due to context size limitations. Phoenix also forms part of the set of 5 games that is highly predictive of the performance on the full Atari suite (Aitchison et al., 2023). We use the Arcade Learning Environment (Bellemare et al., 2013) version, and for expert demonstrations we use the Gato training data (Reed et al., 2022). We use the original (i.e., not down-sampled or grayscale) images as observations (see Fig. A1).

**Chess**   We evaluate models against the *weakest-possible* version of Stockfish 16 (Romstad et al., 2008), i.e., level 0 (which corresponds to an Elo of approx. 1320). We further weaken this version by evaluating only 1 node. We generate expert demonstrations with the strongest (default) version of Stockfish, i.e., level 20 with a time limit of 50ms per move as the agent (note that the opponent, i.e., the "environment", remains fixed as the weakest version of Stockfish). We evaluate four different state representations (see Fig. A3): (i) a 2D ASCII encoding of the board, (ii) the Forsyth–Edwards Notation (FEN), which encodes the board and very limited

historical information as a string, (iii) the Portable Game Notation (PGN), a plain text format for recording chess games via the move history given in algebraic chess notation, and (iv) an RGB image of the board. We always represent actions via the algebraic chess notation. To ensure variability in the demonstration and evaluation episodes, we use the openings from the Encyclopedia of Chess Openings (Matanović, 1978), which we randomly sample without replacement. We play all games for at most 100 steps (terminating early in case of a win/draw/loss; the average number of steps per game is 38) and assign a reward of 1 to a win, 0 to a draw, $-1$ to a loss, and 0 to all other states.

**Crossword**   We create a large collection of $7 \times 7$ crosswords using the genxword crossword generator (Whitlock, 2011) and a list of 55 189 clues with the *lowest difficulty rating* collected by Matthew Ginsberg (individual clues may appear in multiple crosswords with low probability). Each episode is a distinct crossword represented as an ASCII crossword grid followed by two lists of clues, one for the "Across" words and one for the "Down" words (see Fig. A4). A valid action consists of either "A" (for across) or "D" (for down), followed by the word's index and the word itself. We assign a reward of 1 to a correct word, 0 to an incorrect word with correct length or to a correct word that has been already been placed, and we terminate the episode with a reward of $-1$ for an incorrect word. We evaluate 25 steps (which is sufficient to place the approx. 10 words per crossword on average) and terminate early if all words are filled. We generate the expert demonstrations by simply outputting the solution for each clue one by one in random order.

**DM Control – Cheetah Run**   We use the Cheetah Run task from the DM Control Suite (Tassa et al., 2018). We represent observations as string in the style of a Python dictionary of position and velocity vectors with individual values between $-1$ and 1 (see Fig. A5). Each episode begins in a new, randomly initialized state. We only evaluate the first 100 steps of an episode (due to context length limits). As for Atari, we use the Gato training data (Reed et al., 2022) to create expert demonstrations (details in Appendix B.2).

**Grid World**   We consider a $12 \times 12$ grid world with walls, effectively yielding a $10 \times 10$ grid, with a single player and a single target (no obstacles). Actions are up, down, left, and right, and since the grid is fully observable, memorization is not required. We evaluate three different state representations (see Fig. A6): (i) an ASCII encoding of the 2D grid, (ii) the player and target coordinate tuples provided as plain text, and (iii) the RGB image of the grid. The reward is 1 if the player reaches the target and 0 otherwise. Episodes run for a maximum of 25 steps (reaching any target from any initial position in a $10 \times 10$ grid requires at most 18 steps). We randomly initialize the player/target locations for every

episode. The expert demonstrations correspond to one of the shortest paths between the player and the target.

**Tic-Tac-Toe** We play tic-tac-toe against a *random* policy that picks an empty spot uniformly at random. For demonstration episodes the expert uses minimax search (i.e., it plays an optimal action), while the environment remains fixed as a random policy. We evaluate two state representations (see Fig. A2): (i) an ASCII encoding of the 2D grid, and (ii) the RGB image of the grid. To ensure variability, each game starts from an opening state (similar to chess openings), which we draw uniformly across all initial game states (as a result, even the optimal minimax strategy can only win $85\%$ of the games and draws the rest). We assign a reward of 1 to a win, 0 to a draw, $-1$ to a loss, and 0 else.

## 2.3. Prompt

Our prompt consists of two main parts: (i) the expert demonstration episodes (Listing 1), and (ii) the trajectory of the evaluation episode (including the episode's previous actions and environment states; see Listing 2). We use a generic decision-making zero-shot preamble at the beginning of (i), and a short separator prompt at the beginning of (ii) (see Fig. 1 and Listings 1 and 2). The expert demonstrations are fixed across an evaluation episode, but resampled across episodes, while the current trajectory starts with a single initial state and grows as more state-action pairs are observed (up to a maximum of 100 steps). We do not include an environment task description (e.g., the rules of tic-tac-toe) in the preamble. We may, however, show the available legal actions, which depend on the environment, in each step at the end of (ii) (see Listing 2), depending on whether it is beneficial per model and task (see our ablations in Appendix C.4). Similarly, we may use a chain-of-thought (Wei et al., 2022) style prompt at the very end of (ii) asking the model to provide a reasoning before proposing an action (see Listing 2), again, depending on whether it increases performance in the ablations (Appendix C.4). We make both of these decisions per model and task, i.e., the same model may use chain-of-thought for one task but not another.

## 2.4. Evaluation Protocol

In every evaluation step, we condition the model on the current context, upon which it generates an action that we feed to the environment. We then concatenate the resulting next environment state (prepended with the action that produced it) to the growing context. We ablated whether to show the model's previous actions in the evaluation trajectory (see Appendix C.4.1) and found that it mostly improves performance, so we always include them. If a model fails to generate a legal action, we uniformly sample one of the legal actions (we visualize the percentage of illegal actions in Appendix C.3). Since the models have different maximum

context lengths (and different text/image tokenizers), the maximum number of demonstration episodes that fit into the context depends on the model, the task, and the state representation format. For example, for a given model, we may only by able to use 16 demonstration episodes with RGB observations but up to 256 ASCII demonstration episodes.

We always evaluate 100 episodes with different initial conditions (each episode is evaluated individually) and report the average score. The maximum episode duration is 100 steps and the score per episode is the cumulative reward over all steps (we never show reward information to the models). For each evaluation episode we uniformly subsample (without replacement) the demonstration episodes (for the frozen part of the prompt) from a precomputed pool of up to 1000 distinct demonstrations. We ensure that all evaluation episodes have initial states that differ from the demonstration episodes (except for the replay control experiments in Appendix C.2). We only perform very minimal postprocessing (see Section 2.1) of the LM generations to obain the action and (deliberately) reject semantically correct actions that are wrongly formatted, as matching the action format is an important aspect of our imitation learning benchmark.

## 3. Results

We now present our comprehensive empirical evaluation of the (closed-source) frontier models on our benchmark for interactive decision-making with long multimodal context. We investigate how performance changes when presenting more demonstration episodes in the context (see Section 2.2 for a task overview and Appendix B.2 for details and illustrations). For each model and task, we first ablate whether to use chain-of-thought prompting and whether to show the legal actions in the prompt (results in Appendix C.4). We use one demonstration episode (i.e., still many individual demonstration steps) for these ablations, as a compromise between lower computational demands and being representative of the in-context learning setting. Due to the very low monthly rate limits of the Anthropic API (Anthropic, 2024d), we do not sweep over the number of demonstrations for Claude 3.5 Sonnet and only run the ablation. We also do not evaluate it on Atari since it can only process 100 images at once (a single demonstration episode hits that limit).

### 3.1. Best Scores Per Model/Task

Fig. 2 shows the highest overall score per task and model across all settings (number of demonstrations, observation format, showing legal actions, chain-of-thought prompting). Different models thus may use different settings on the same task, and the same model may use different settings across tasks. In Section 3.2, we will keep the observation format and number of demonstrations constant across all models per data point. As stated above, results for Claude 3.5 Son-

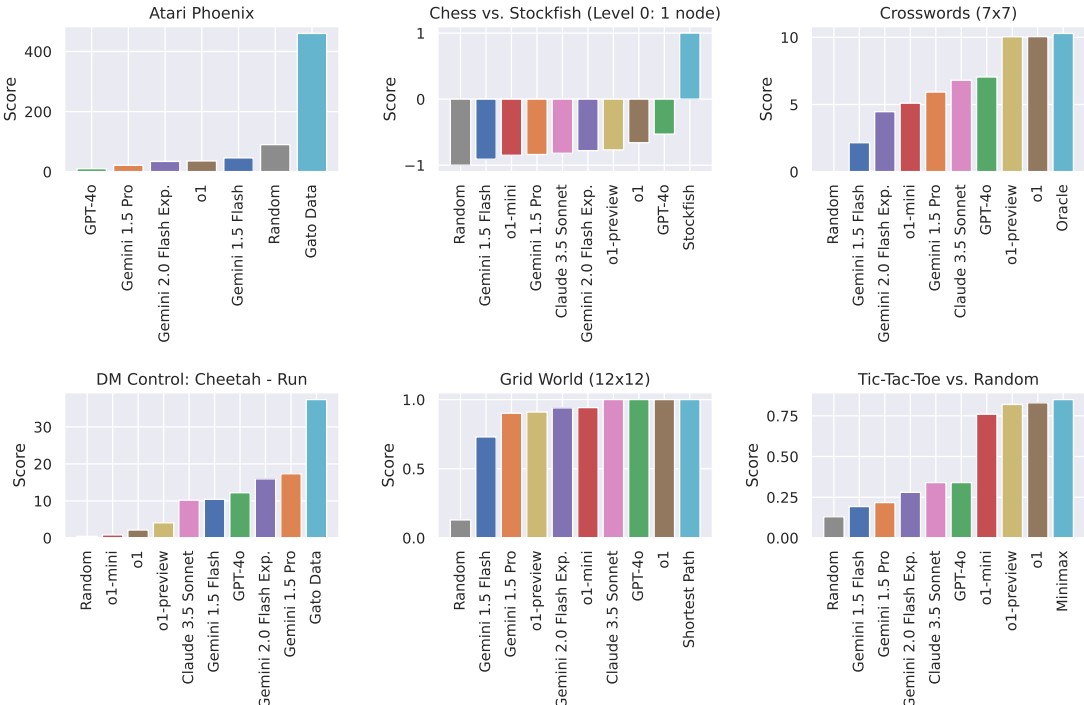

*Figure 2.* Best scores per model and task across all observation formats, numbers of demonstration episodes, and ablations (chain-of-thought, showing legal actions). Accordingly, different bars in a panel may be based on different settings. The expert policy (which produced the demonstrations) is an upper baseline. The lower baseline randomly selects a legal action at each step. Claude 3.5 Sonnet, o1-mini, and o1-preview cannot be evaluated on Atari – Phoenix because they cannot process (enough, for Claude 3.5 Sonnet) images.

net are from the ablation (i.e., 1 demonstration episode). Fig. 2 shows that models often struggle to match expert scores — even in their best setting. Exceptions are grid worlds, which most models largely solve, tic-tac-toe, where the o1 models achieve near-optimal score (against a random opponent), and crosswords, which o1-preview and o1 almost solve. On the other hand, all models always outperform the random action baseline except for Atari – Phoenix. LMs tend to repeat actions (see Fig. A14 for a detailed analysis), and in Phoenix holding down the firing button without releasing results in a single shot (no auto-fire). In contrast, random actions produce a fair amount of (successful) shots.

### 3.2. In-Context Imitation Learning

We now investigate the in-context imitation learning capabilities of today's frontier models, by varying the number of demonstration episodes from zero-, to few-, and many-shot evaluations. The largest number of demonstrations depends on the models' context sizes, the task, and the tasks' observation formats (images require more tokens than text). We omit Claude 3.5 Sonnet due to its low monthly token limit.

**Atari – Phoenix**   Fig. 3 shows that on Atari – Phoenix all models except o1 improve slightly by having one demonstration episode, as opposed none. However, more demonstra-

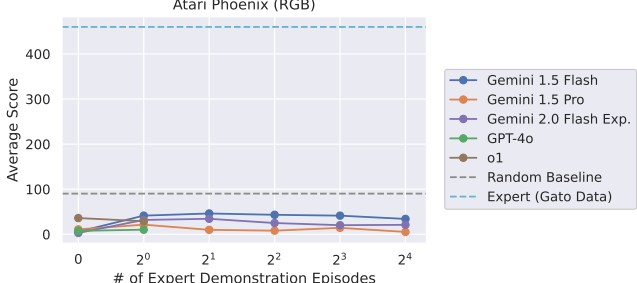

*Figure 3.* In-context imitation learning on Atari – Phoenix (RGB observations). Almost all models benefit (mildly) from one demonstration episode, but not from more (GPT-4o and o1 cannot fit multiple demonstration episodes in the context). While no model outperforms the random baseline, Gemini 1.5 Flash performs best.

tions (only possible for the Gemini models) do not improve the performance further. All models struggle to outperform the random action baseline because of their tendency to repeat actions (and thus fire very little, since Phoenix has no auto-fire). Table A1 contains the ablation results. Models rarely output illegal actions (see Fig. A15), i.e., they reliably (re)produce the correct action format. Atari – Phoenix is arguably the hardest task in our benchmark, since it only comes with image observations (no text representation), and has very large demands w.r.t. context size (high frame rate,

somewhat high resolution images, etc.). Despite LMs' massive scale, playing Atari games well (at least by naively feeding raw images to the context) seems currently beyond reach, both in terms of capabilities, but also w.r.t. context size and compute demands (cf. Waytowich et al. (2024)).

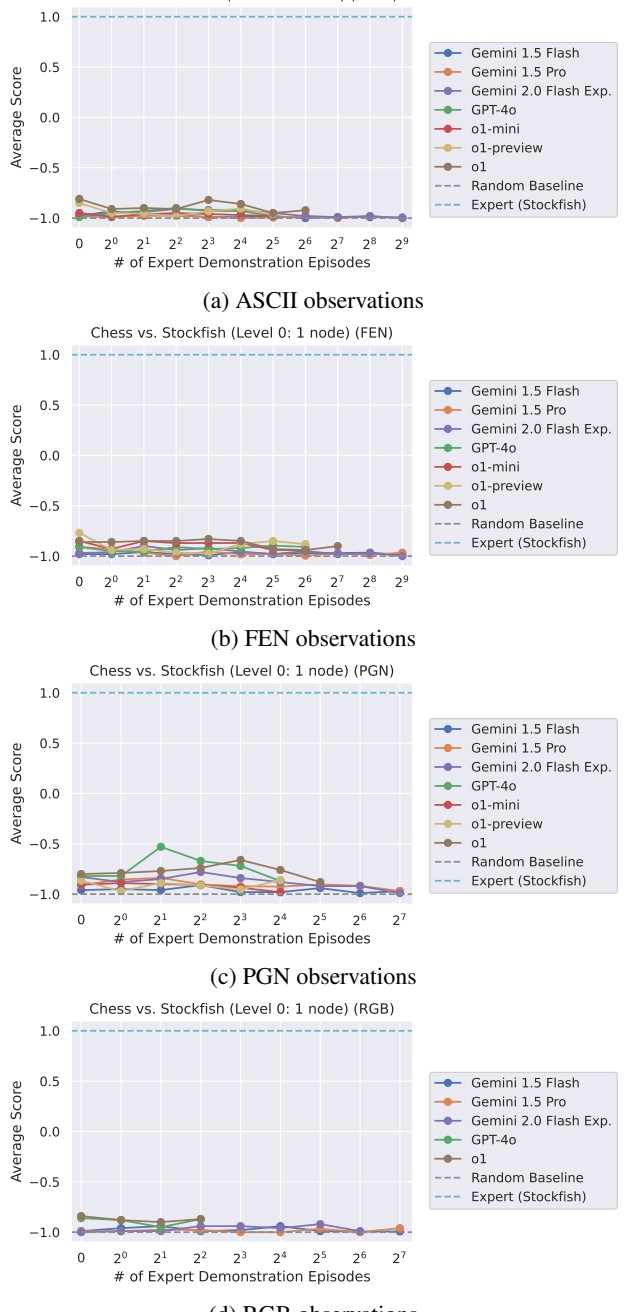

(a) ASCII observations

(b) FEN observations

(c) PGN observations

(d) RGB observations

Figure 4. In-context imitation learning on chess against the weakest variant of Stockfish (level 0, $\approx 1300$ Elo), further restricted to one node. The models almost always lose (i.e., score $-1$) and do not benefit from more demonstrations. The PGN observations enable the best results, in particular for GPT-4o, which performs best but still loses majority of games against this weak opponent.

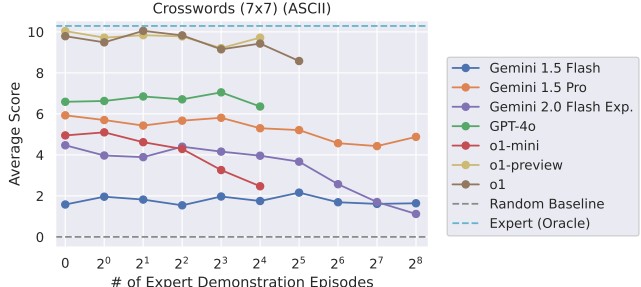

Figure 5. In-context imitation learning on $7 \times 7$ crossword puzzles (using clues with the simplest rating) with ASCII observations. The performance of most models is largely unaffected by the number of expert demonstration episodes. o1-preview and o1 solve most crosswords, while other models struggle to varying degrees.

**Crosswords** Fig. 5 shows LMs' performance on crosswords with simple clues (as rated by Matthew Ginsberg). While the models achieve very different overall scores, their individual performance is largely independent of the number of demonstrations (except for o1-mini and Gemini 2.0 Flash Experimental, which degrade with more demonstrations). Overall, o1-preview and o1 perform best, almost completely solving all the (simple) crosswords. Fig. A17 shows that the number of illegal actions (i.e., where models either suggest a word of incorrect length or fail to respect the "Across" vs. "Down" format) is quite high for all models and roughly inversely proportional to their puzzle-solving competence. We use chain-of-thought prompting for some but not all models (see our ablation results in Table A5), but we never show the legal actions (which would be a unreasonably long list of words with correct length).

**Chess** Fig. 4 shows LMs' chess-playing performance against the weakest version of Stockfish 16 (Romstad et al., 2008) (i.e., level 0, $\approx 1300$ Elo), further restricted to only evaluating a single node. We investigate four observation formats: ASCII, FEN, PGN, and RGB (o1-mini and o1-preview cannot process images). Showing more demonstrations has little effect on performance, and models rarely manage to beat (score 1) or draw (score 0) against this very weak opponent. The results show that playing chess (without any scaffolding or fine-tuning) is still out of reach for current LMs. Fig. A16 reveals that the models often output illegal actions (which also does not improve with more demonstrations) — even though the legal actions are provided in the prompt (cf. the ablations in Tables A2 to A4).

**Tic-Tac-Toe** Fig. 6 shows the results for playing tic-tac-toe against a random-action opponent. Since the demonstration and evaluation episodes start from different initial states, episodes begin with partially filled boards, which cannot always be won (i.e., optimal score $< 1$). The o1 models reach expert performance with ASCII observations

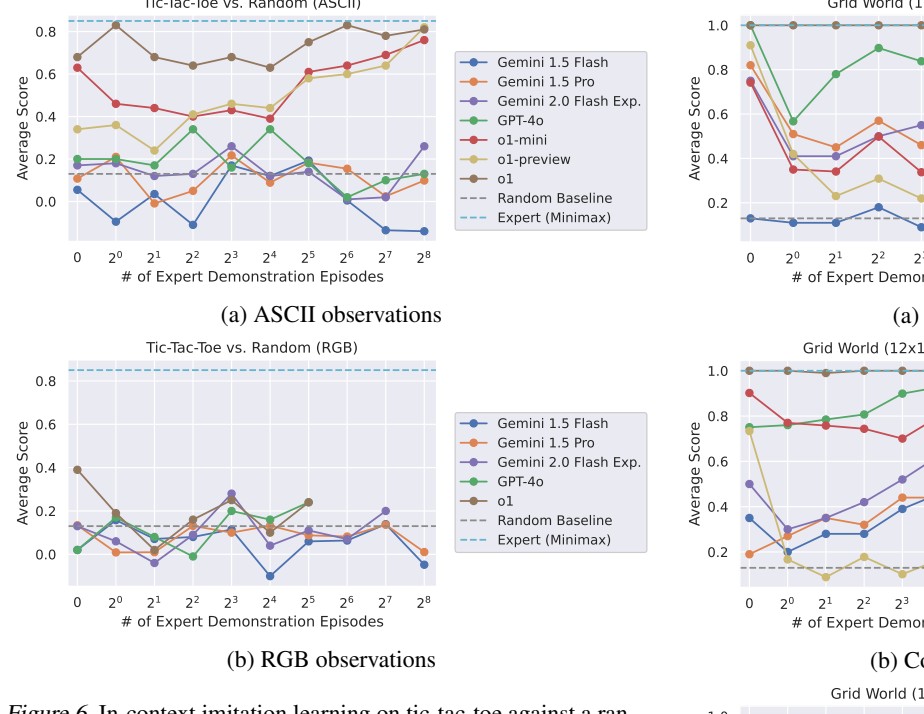

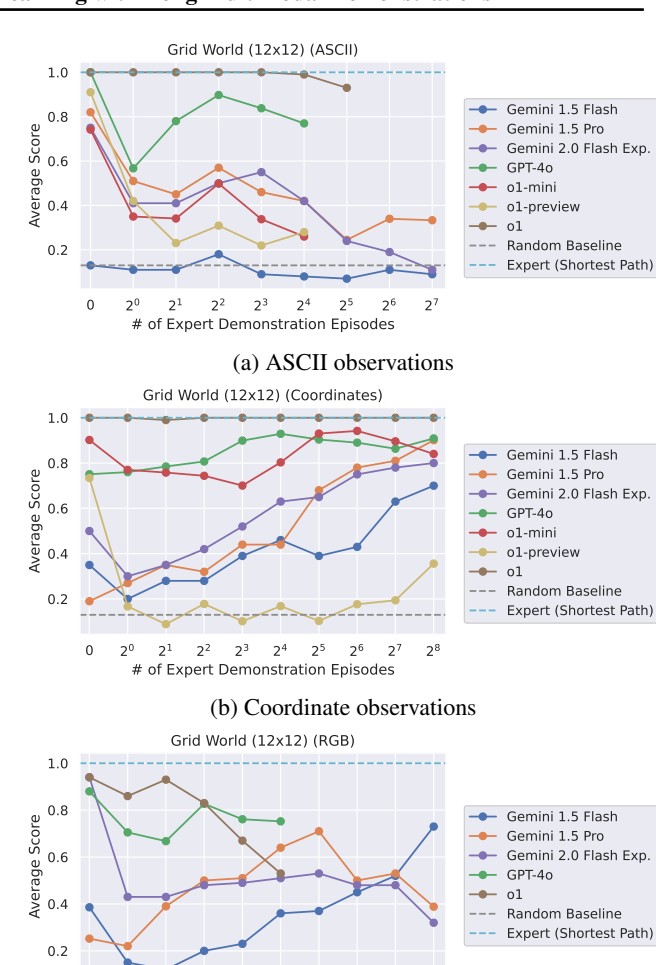

*Figure 6.* In-context imitation learning on tic-tac-toe against a random action adversary. Apart from the o1 models on ASCII observations, all models struggle to play better than a random baseline. o1-mini and o1-preview improve with more demonstrations and both reach expert performance at 256 demonstration episodes.

and show signs of in-context learning. The other models struggle to outperform the random action baseline (as does o1 with RGB observations). Fig. A20 shows that the models (apart from Gemini 1.5 Pro on RGB) generate few illegal actions, implying that the models' weak performance is caused by outputting suboptimal rather than illegal actions.

**Grid World**   Fig. 7 shows how LMs perform on the task of navigating a simple grid world. The Gemini models steadily improve with more demonstrations for coordinate and RGB image (1.5 models only) observations, demonstrating strong in-context learning with very long contexts. While GPT-4o, o1-mini, and o1-preview do not benefit from more demonstrations, o1 performs very well in almost all settings. On ASCII observations, most models (except o1 and Gemini 1.5 Flash) deteriorate with more demonstrations. Fig. A19 shows the illegal actions for each model, revealing that o1-preview's performance drops on ASCII and coordinate observations strongly correlate with the number of illegal actions. All other models rarely generate illegal actions after one demonstration episode. Overall, grid world is the easiest task in our benchmark where all models perform quite well (some almost optimally) with the right combination of state representation and number of demonstrations.

*Figure 7.* In-context imitation learning for navigating to a target in a $12 \times 12$ grid world (see Fig. A6) using the commands: up, down, left, right. On ASCII, most models deteriorate with more demonstrations. For coordinate observations (player and target tuple) and RGB images, Gemini improves with more demonstrations (except Gemini 2.0 Flash Experimental on RGB), indicating in-context learning across a very long context. GPT-4o (and o1 on RGB) shows no such a trend but already achieves high zero- and few-shot performance (o1 is near-perfect on ASCII and coordinates).

**DM Control – Cheetah Run**   Fig. 8 shows LMs' ability to control a simulated (half) cheetah from DM Control (Tassa et al., 2018). We encode observations and actions as strings (see Fig. A5). For all models, except o1-mini, a single demonstration episode is helpful but more than two demonstrations degrade performance. The o1 models struggle to significantly outperform the random action baseline. Gemini 1.5 Pro and Gemini 2.0 Flash Experimental achieve the highest score, reaching roughly half the expert score. Except for o1-mini, all models mostly generate legal actions (see Fig. A18) — even without any demonstration episodes

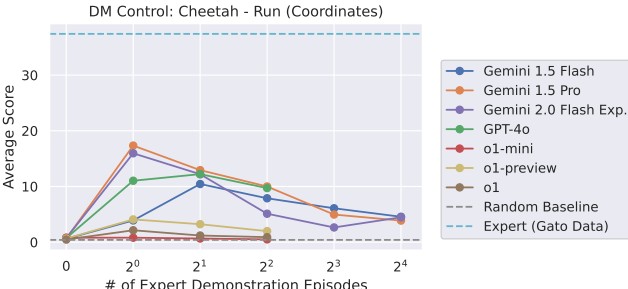

*Figure 8.* In-context imitation learning on the cheetah run task from the DM Control suite using position/velocity vector observations encoded as text. Gemini 1.5 Pro and Gemini 2.0 Flash Experimental perform best with a single demonstration episode, GPT-4o and Gemini 1.5 Flash with two demonstration episodes, with more demonstrations degrading performance. Gemini 1.5 Pro achieves the highest performance, roughly half the expert score.

(the action format can be inferred from the past actions in the evaluation trajectory since we randomly sample a legal action if the model generates an illegal one). Therefore, the poor performance in the zero-shot regime (i.e., 0 demonstration episodes) cannot be explained by the models' potential lack of knowledge of the action format. Instead, models (with the exception of o1-mini) manage to learn a non-trivial policy from one or two demonstration episodes (but fail to leverage more episodes to further improve performance).

## 4. Discussion & Related Work

Many large-scale benchmarks have been developed to test the general capabilities of frontier LMs, including the Chatbot Arena (Chiang et al., 2024) or LiveBench (White et al., 2025). Other benchmarks, such as Frontier-Math (Glazer et al., 2024), TaskBench (Shen et al., 2024), GameBench (Costarelli et al., 2024), and Atari-GPT (Way-towich et al., 2024), specifically investigate the reasoning or interactive decision-making capabilities of (L)LMs. Most closely related to our work, and published in parallel, the BALROG (Paglieri et al., 2025) benchmark evaluates frontier LMs' zero-shot reasoning on multimodal decision-making tasks in five game environments from Baby AI to NetHack. Our work complements these benchmarks by focusing on in-context imitation learning with long multimodal context, filling a gap in the current literature. Our benchmark is currently in its "easiest" form (e.g., grid world without obstacles) and can easily be made more challenging.

There are a few reasons to be optimistic about in-context imitation learning with long-context models. Many demonstrations should, in principle, improve performance over few-shot learning (as shown by Agarwal et al. (2024); Jiang et al. (2024) on non-interactive tasks), and modern LMs have long enough contexts to test this at scale. Additionally,

pretrained LMs have a fairly general ability to recognize and imitate algorithmic patterns in their context (Mirchandani et al., 2023), which is in line with the memory-based meta-learning view on (universal) in-context prediction (Ortega et al., 2019; Grau-Moya et al., 2024). If universal enough, models would, with enough observations, recognize and correctly continue environment-agent interaction patterns. See Appendix A for a discussion of additional related work.

While we find some cases of steady in-context learning, in the majority of cases LMs' performance is largely independent of the number of demonstrations. It is unclear whether in-context imitation learning is not well suited to communicate desired agentic behavior (i.e., models do not "understand" the task purely from demonstrations), or whether models have difficulty to effectively use a dense long context. To provide additional insight we perform a "replay" control experiment, where a single episode is shown in context, and the same exact episode is evaluated, such that models only need to "copy" actions from the demonstration episode. We find that models (except o1-mini) perform well in this control experiment on all tasks (see Appendix C.2).

**Limitations** We perform an evaluation via closed-source APIs and thus have little control over how the data is processed and fed to the underlying models. Since models behind the APIs can be updated at any time, it is possible that our results may not be quantitatively reproducible soon after publishing this manuscript. Despite our best efforts in evaluating different prompt formats, we cannot rule out that even small changes to the prompt could lead to better results. Accordingly, our current results are a lower bound on the models' performance. We can also not guarantee that our observation formats do not cause tokenization issues across all APIs (e.g., a loss of structure for a 2D grid in ASCII). Finally, while our environments are simpler than complex real-world scenarios like robotics, they require the same fundamental agentic capabilities (long-context multimodal understanding, in-context imitation learning) that are often tested in robotics research and likely necessary for real-world success. Accordingly, our LMAct benchmark serves as a controlled testbed to evaluate these core skills, diagnose current model limitations, and guide research toward more generally capable agents. Although direct transfer is beyond this paper's scope, findings on our benchmark can inform future work addressing that challenge.

**Future Work** For Cheetah Run (Fig. 8) and grid world navigation with ASCII demonstrations (Fig. 7) we observe that the models' performance deteriorates with increasing numbers of expert demonstrations, a phenomenon we refer to as "in-context interference". For both tasks our analysis of the percentage of illegal actions (Figs. A18 and A19) suggests the problem is not due to an increased number of illegal actions, but, instead, due to increasingly suboptimal actions. We have conducted a set of initial investiga-

tions into these failure modes (see Appendices C.2 to C.4 and Fig. A14), but a definitive answer to what causes in-context interference would require a thorough investigation, and, therefore, presents an interesting direction for future work. Another promising avenue for future work is to focus on models capable of general in-context reinforcement learning, which is a bit different than our in-context imitation setting (in principle, all our tasks could easily be extended by providing additional reward observations). It also seems plausible that pretraining or finetuning with data from interactive decision-making tasks, and, in particular, in-context imitation of an expert policy, would be quite effective. Finally, we also think that evaluating LMs' performance on partially observable tasks presents an interesting direction for future research (we primarily investigate fully observable tasks). Partially observable tasks require consistent integration of information across several, potentially non-adjacent, time steps, which is a great test of a model's ability to densely attend to the information in the context. However, under partial observability, there is a theoretical problem of self-delusion in imitation of an expert that has hidden information (e.g., the hidden belief state of the agent; see Ortega et al. (2021)). We want to keep our benchmark free from these complicating issues and think that the right time to move to such harder tasks is when frontier models easily solve simple, fully observable tasks at expert level (we are currently at beginner level). Nevertheless, we believe there is great value in benchmarks on interactive decision-making tasks under partial observability (perhaps better suited for in-context reinforcement learning, which avoids the self-delusion problem), and refer to, e.g., the BALROG benchmark (Paglieri et al., 2025).

## 5. Conclusion

We evaluated the multimodal in-context imitation learning capabilities of some of the world's most advanced AI models on interactive decision-making tasks — tasks that are simple for humans but challenging for state-of-the-art LMs. Our results show that, even with hundreds of demonstration episodes, context lengths of up to one million tokens, and thousands of output tokens, models often struggle to reach expert performance, thereby failing to translate their (factual) knowledge about the tasks' solution strategies into effective decision-making. Solving this problem will be crucial for the next generational leap in LM capabilities towards general agents. Despite our focus on in-context imitation learning, we believe it will be interesting to compare against other methods, including fine-tuning, retrieval-based methods, reward-conditioning, etc. Our open-source benchmark (`https://github.com/google-deepmind/lm_act`) serves as a yard stick to measure progress toward that goal, and we are excited to see which innovations will be needed to solve all our simple tasks.

## Impact Statement

The (partial) automation of intellectual labor will have significant societal and socioeconomic impact, both positive and negative. One current hurdle is the automation of general interactive decision-making tasks, as our benchmark demonstrates. If solved, this would lead to more autonomous AI systems and agents that could facilitate or even partly take over intellectual labor across a broad range of applications and domains. We believe this vision warrants both enthusiasm and caution, and the scientific community and developers of AI technology bear responsibility in raising awareness of potential negative socioeconomic outcomes, and in helping develop potential mitigations. Having said that, our work is a benchmark that reports the current state of affairs (showing the limitations of current frontier LMs) and serves as one yard stick to measure future innovations and progress against, but does not propose innovations to improve capabilities of AI systems. While our benchmark may contribute indirectly by helping develop more capable AI systems faster, we believe this is far outweighed by the benefit of having a clearer picture of current capabilities and progress provided by our benchmark.

## Acknowledgments

We thank Anna Mitenkova, Dipanjan Das, Jordi Grau-Moya, Joel Veness, Kate Olszewska, Li Kevin Wenliang, Laurent Orseau, Marcus Hutter, Matthew Aitchison, Matthew Ginsberg, Minmin Chen, Orhan Firat, Satinder Baveja, Shaobo Hou, Zhengdong Wang, and the anonymous reviewers for their helpful feedback and insightful discussions.

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

# A. Related Work

The emergence of strong in-context learning capabilities with increasing model and training data scale in LLMs was first observed in the GPT-2 paper (Radford et al., 2019) and even gave the title to the GPT-3 paper (Brown et al., 2020). Soon after, the recipe of next-token prediction at scale was applied to build agents from large sequential predictors, resulting, e.g., in the decision (pretrained) transformer (Chen et al., 2021; Lee et al., 2023), or the generalist agent GATO (Reed et al., 2022), and its more recent open source variant JAT (Gallouédec et al., 2024). While the empirical results were certainly surprising at the time, at least in theory, in-context learning must necessarily arise as a core feature of a sequential predictor trained to minimize next token log loss over an implicit meta distribution of data (Ortega et al., 2019; Mikulik et al., 2020; Genewein et al., 2023), and could, in principle, even lead to universal in-context predictors (Grau-Moya et al., 2024). An explicit application of this memory-based meta-learning principle at scale is the "adaptive agent" from Bauer et al. (2023), which shows that an embodied agent in a 3D environment can adapt to novel task instances on human timescales (i.e., single- or low double digit numbers of interaction episodes) purely in context and across a vast set of tasks. The SIMA Team et al. (2024) conducted another impressive large-scale application of training and fine-tuning a complex vision-language agent across a large set of environments, using instruction conditioning together with in-context learning.

Instead of pretraining separate models for general decision-making, many researchers have also attempted to directly use the knowledge and reasoning capabilities of LLMs and VLMs for building agents that interact with an environment (see Xi et al. (2025) for a 2023 survey of LLM-based agents) — either by fine-tuning (Li et al., 2022) or purely via in-context demonstrations (Palo & Johns, 2024). In both cases, two open question are: (i) how to best represent environment observations as tokens, and (ii) how to best elicit the decision-making capabilities of pretrained LMs. Mirchandani et al. (2023) find that pretrained LLMs are "general pattern machines" that can learn to complete complex token sequences via in-context learning, including agent-environment interactions. They even find that in many cases, randomly swapping the alphabet does not have significant impact, suggesting that LLMs may be able to deal with many different ways of translating observations into tokens. Perhaps more important is the question whether observations should be state-action sequences (as in imitation learning and our work) or whether they should include rewards (for reward conditioning or in-context RL as in Mirchandani et al. (2023) and Raparthy et al. (2024)). While this is still unclear for pretrained LMs, Ruoss et al. (2024) find that, when training a large transformer to play chess, performance is roughly the same for imitation learning compared to learning to predict state or action values (as long as the amount of data for all three variants is equal). Schultz et al. (2025) extend these results on chess to LLMs by distilling the search proceedure into the model by linearizing the search trees. Ma et al. (2025) find that, when prompted appropriately, a pretrained VLM (Gemini 1.5 Pro) can produce good value estimates for real-world robotic tasks. The second open problem, how to best prompt LMs for decision-making, is a very active research area (Wenliang et al., 2025; Genewein et al., 2025), with a lot of focus on designing or learning zero-shot prompts, such as the famous "Let's think about this step by step." (Kojima et al., 2022) and "Take a deep breath and work on this problem step-by-step." (Yang et al., 2024), which has led to many chain-of-thought prompting schemes (Wei et al., 2022). Besides better zero-shot prompts, advanced sampling and prompt optimization schemes have been explored, such as iterative prompt refinement where an agent starts with some demonstrations in the context, then interacts with the environment, and potentially replaces a lower performing episode with a higher performing one (Mirchandani et al., 2023; Brooks et al., 2023).

With the recent availability of long context models, a third possibility compared to improving zero-shot or few-shot prompts has emerged: many-shot in-context learning, i.e., prompting with many demonstrations, on the order of having a full small dataset in the context. Both Agarwal et al. (2024) and Jiang et al. (2024) show that many-shot prompts (hundreds or thousands of examples in the prompt) improve pretrained LM performance over few-shot prompts in non-interactive tasks. Our paper explores the same direction, with potentially hundreds of full *episodes* in the context, for interactive decision-making tasks. At the time of writing, querying long context models with many tokens comes with high computational cost, but it would be interesting to investigate how specialized models that were specifically developed for long-context tasks (Bulatov et al., 2022; Cherepanov et al., 2024) would fare on our benchmark. An alternative could be to virtually extend the context via retrieval based methods, e.g., REGENT (Sridhar et al., 2024), which trains a retrieval based agent. The retrieval problem is currently not fully solved (as also pointed out in Paglieri et al. (2025)), and today's largest state-of-the-art LMs do not offer retrieval via their APIs and/or have not been trained to perform retrieval. Accordingly, placing all the data in the context, as in our work, allows estimating LM performance independent of whether retrieval works well. Our results thus serve as an upper baseline to calibrate retrieval based methods against.

Another line of work has investigated in-context learning for control (Duan et al., 2017; Xu et al., 2022; Fu et al., 2024; Fang et al., 2024; Yin et al., 2024; Wang et al., 2024). While these methods focus on developing novel agents with strong in-context learning capabilities, our paper's primary goal is to benchmark the current state of existing, general-purpose

frontier LMs on such tasks. Understanding how to best leverage insights from specialized agent research to enhance these large, pre-trained models remains an open question, which underscores the importance of establishing clear benchmarks like LMAct.

Comparing capabilities of LMs that span many tasks and domains requires large-scale benchmarks, such as the Chatbot Arena (a.k.a. LMSys, Chiang et al. (2024)) or LiveBench (White et al., 2025). Other benchmarks, such as FrontierMath (Glazer et al., 2024), Taskbench (Shen et al., 2024), Gamebench (Costarelli et al., 2024), and BABILong (Kuratov et al., 2024), specifically investigate the reasoning or interactive decision-making capabilities of LLMs. Closely related to these and to our work, and published in parallel to our work, the BALROG (Paglieri et al., 2025) benchmark evaluates state-of-the-art LMs (the same as in our work with the exception of o1-mini and o1-preview and the addition of the Llama 3 models, which we do not evaluate) on multimodal reasoning and decision-making tasks by using a set of 5 increasingly harder game environments from Baby AI to Nethack. Like our work, the authors find that state-of-the-art LMs struggle significantly in challenging game environments. The authors also make the observation that models arguably possess a lot of knowledge about their tasks when queried appropriately, but that state-of-the-art LMs have a "knowing-doing" gap. Unlike our work, the BALROG paper performs zero-shot evaluation and lists few-shot and many-shot evaluations as an important open research question (the released codebase supports few-shot evaluations, but these evaluations have not been performed at the time of writing). Similar to BALROG, Waytowich et al. (2024) evaluate the zero-shot game-playing capabilities of frontier LLMs (GPT-4V Turbo, GPT-4o, Gemini 1.5 Flash, and Claude 3 Haiku) on 8 different Atari games (but not the Phoenix game that we consider). Compared to these previous benchmarks, ours is the only one that puts the emphasis on imitation learning with long context in multimodal interactive environments — a regime that pushes against the limits of modern LMs' in-context learning and reasoning capabilities, both from an engineering and a capabilities perspective. Furthermore, our benchmark covers the zero-shot, few-shot, and many-shot setting in a unified and thus easily comparable evaluation.

## B. Experimental Details

In this section, we provide additional details on our experimental setup.

### B.1. Models

**Claude 3.5 Sonnet**    Claude 3.5 Sonnet has a maximum context of 200k tokens, but can only process 100 images at a time. In our setting, a single episode can already consist of up to 100 images. As a result, we can, e.g., not evaluate in-context learning for Claude 3.5 Sonnet on Atari. Claude 3.5 Sonnet uses approximately $(\text{width px} \cdot \text{height px})/750$ tokens per image (Anthropic, 2024e). Finally, note that the monthly token limits for Claude 3.5 Sonnet are very low compared to the other models ($5000 spend limit per month (Anthropic, 2024d)), which is why we can only conduct a somewhat limited evaluation of this model.

**Gemini 1.5 Flash, Gemini 1.5 Pro, and Gemini 2.0 Flash Experimental**    The Gemini 1.5 models have a maximum context window of 10 million tokens. In practice, we restrict ourselves to 1 million tokens due to the prohibitive cost of evaluating even longer prompts. The Gemini API considers images to be of fixed size, meaning that they consume a fixed number of tokens (currently 258), regardless of their file size (Google, 2024). Gemini 2.0 Flash Experimental supports up to 1M input tokens and 8k output tokens (Google DeepMind, 2024a).

**GPT-4o**    The GPT-4o model has a context length of 128k tokens and can process up to 250 images per prompt (OpenAI, 2024b). We use the "auto" resolution setting to process the images, which automatically determines whether to use a "low" or "high" resolution based on the image size (OpenAI, 2024e). In the "low" resolution mode, the model represents an image with 85 tokens. In the "high" resolution mode, the model first consumes the low-resolution image (using 85 tokens) and then creates additional detailed crops of the image using 170 tokens per $512\text{px} \cdot 512\text{px}$ tile to cover the whole image.

**o1-mini, o1-preview, and o1**    o1-mini and o1-preview have a context window of 128k tokens with up to 65k output tokens but cannot process images (OpenAI, 2024d) so we only evaluate them in the text-based environments (e.g., not on Atari). The o1 model has a context window of 200k tokens and up to 100k output tokens.

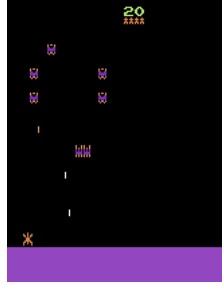

*Figure A1.* An RGB observation for the Phoenix game for Atari 2600.

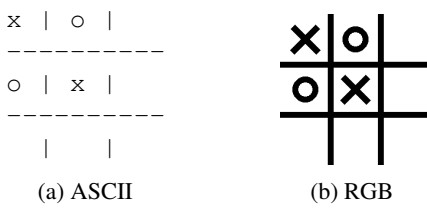

(a) ASCII      (b) RGB

*Figure A2.* Sample observations for each state representation format from our tic-tac-toe environment.

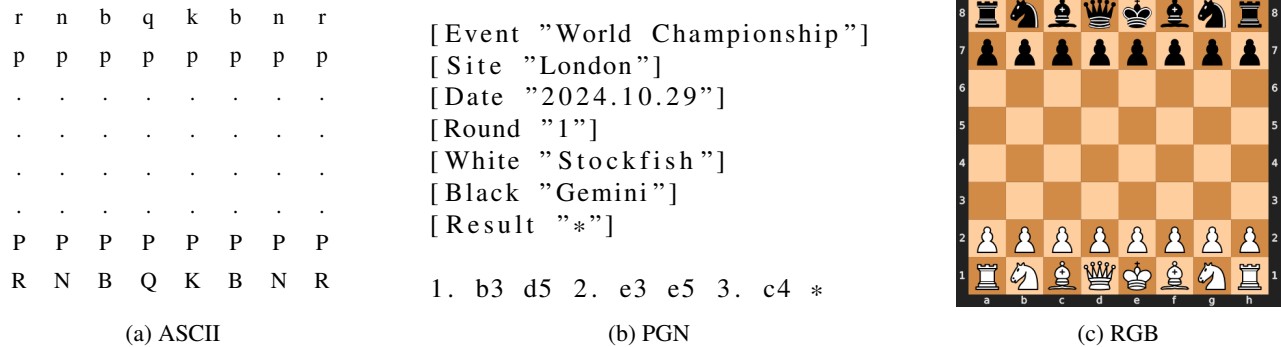

(a) ASCII      (b) PGN      (c) RGB

rnbqkbnr / pppppppp / 8 / 8 / 8 / 8 / PPPPPPPP / RNBQKBNR  w  KQkq − 0  1

(d) FEN

*Figure A3.* Sample observations for each state representation format from our chess environment, all of which we generate with the python-chess library (Fiekas, 2012). Note that, unlike the ASCII, FEN, and RGB, which show the opening board state, the PGN corresponds to a more advanced position to visualize the move list (which would be empty for the opening board state).

## B.2. Environments

**Visualizing State Representations** To evaluate *multimodal* in-context imitation learning, we consider multiple state representations for our environments — which representations exactly depends on the environment. For example, for chess, we evaluate four different formats: (i) ASCII, (ii) FEN, (iii) PGN, and (iv) RGB images. We visualize all formats for each environment in Figs. A1 to A6.

**Atari – Phoenix** Unlike chess - where a superhuman expert policy (i.e., Stockfish) is publicly available, or tic-tac-toe - where the optimal policy can be described with a closed-form algorithm, the best performance on Atari 2600 games is generally obtained by strong reinforcement learning (RL) agents (Mnih et al., 2013; Hessel et al., 2021). However, rather than training an RL policy from scratch (which can be finicky), we make use of the training data corpus of the Gato project (Reed et al., 2022). Concretely, Gato trained a Muesli (Hessel et al., 2021) agent for 200M steps and randomly recorded roughly 20k episodes generated by the agent during training. As a result, the dataset also contains trajectories from the beginning of training where the agent does not yet perform well, so we only consider the last 2048 trajectories (i.e., the final stages of training). We further subsample these trajectories by only keeping the 256 highest-scoring as our expert demonstrations. Since we only evaluate the first 100 steps (i.e., 400 frames with an action repeat of 4), we subsample w.r.t. the cumulative reward in the first 400 frames and not w.r.t. the entire episode. Overall, we obtain a collection of demonstration episodes with a high average return of 459.9 in the first 400 frames. In our experiments we match the Gato setting (Reed et al., 2022), i.e., sticky actions and no uncontrolled random initial no-ops. To ensure variability in the evaluation episodes, we manually perform different numbers of no-ops (based on the random seed) at the beginning of every

```
+----+----+----+----+----+----+----+              +----+----+----+----+----+----+----+
| 01 | ## | ## | 02 |    |    | 03 |              |  S | ## | ## |  G |  A |  M |  E |
+----+----+----+----+----+----+----+              +----+----+----+----+----+----+----+
| 04 |    |    |    | ## | ## |    |              |  A |  N |  T |  E | ## | ## |  A |
+----+----+----+----+----+----+----+              +----+----+----+----+----+----+----+
|    | ## | ## | 05 |    |    |    |              |  K | ## | ## |  M |  U |  I |  R |
+----+----+----+----+----+----+----+              +----+----+----+----+----+----+----+
| 06 | 07 |    | ## | ## | ## |    |              |  E |  R |  A | ## | ## | ## |  N |
+----+----+----+----+----+----+----+              +----+----+----+----+----+----+----+
| ## |    | ## | 08 | 09 |    |    |              | ## |  O | ## |  S |  T |  Y |  E |
+----+----+----+----+----+----+----+              +----+----+----+----+----+----+----+
| 10 |    |    | ## |    | ## |    |              |  E |  W |  E | ## |  E | ## |  S |
+----+----+----+----+----+----+----+              +----+----+----+----+----+----+----+
| ## |    | ## | 11 |    |    |    |              | ## |  S | ## |  M |  A |  R |  T |
+----+----+----+----+----+----+----+              +----+----+----+----+----+----+----+

Across:                                           Across:
 2: Go or Go Fish                                  2: Go or Go Fish
 4: Texas hold 'em stake                           4: Texas hold 'em stake
 5: Sierra Club co-founder John                    5: Sierra Club co-founder John
 6: Cueto stat                                     6: Cueto stat
 8: Eye sore                                       8: Eye sore
10: Shorn female                                  10: Shorn female
11: Store at a gas station                        11: Store at a gas station
Down:                                             Down:
 1: Sushi bar libation                             1: Sushi bar libation
 2: It's set in a setting                          2: It's set in a setting
 3: Sincere, as an apology                         3: Sincere, as an apology
 7: Seating sections                               7: Seating sections
 9: Afternoon event in Chelsea                     9: Afternoon event in Chelsea
```

(a) Initial crossword            (b) Solved crossword

*Figure A4.* Sample observations from our crossword environment. Fig. A4a shows the initial crossword and Fig. A4b shows the solved crossword after all the words have been placed in their corresponding slots. We create the crosswords of size $7 \times 7$ using the genxword crossword generator (Whitlock, 2011) and a list of 55 189 clues collected by Matthew Ginsberg (we only use the clues with the lowest difficulty rating; the full list contains 236 615 clues).

episode before starting the evaluation, which changes the initial state from which the agent has control (since enemies in Phoenix keep moving during this time).

**DM Control – Cheetah Run**    There is generally no closed-form or publicly available expert policy for the tasks from the DM Control suite. Thus, we also leverage the Gato training corpus to generate our expert demonstration episodes for cheetah run. For this task the Gato project trained a D4PG (Barth-Maron et al., 2018) agent, and, like for Atari, we only consider the last 10k episodes (since those correspond to the later stages of training and thus better performance). Ideally we would want to subsample the highest-scoring trajectories as expert demonstrations, but, unfortunately, the underlying MuJoCo (Todorov et al., 2012) physics have changed since the time of the Gato data collection, which means that the rewards in the dataset no longer match the observation-actions pairs in our DM Control environment (i.e., replaying the actions from the same initial state does not yield the same return). Thus, we first replay the actions for every trajectory in our collection (setting the initial state based on the first observation) and use the new returns to subsample the 1000 highest-scoring trajectories (again only considering the cumulative reward in the first 100 steps). Overall, we obtain a collection of demonstration episodes with a high average return of 37.4 in the first 100 steps.

### B.3. Prompts

The frozen part and the dynamic part of the evaluation prompt are illustrated in Listing 1 and Listing 2, respectively.

## C. Additional Results

In this section, we present additional results and ablations from our experimental evaluation.

### C.1. Ablating the Maximum Sample Length

The o1 family of models tends to generate (long) internal "reasoning traces" before returning an output. Thus, if the maximum sample length is not large enough, these models may not have enough "reasoning tokens" and therefore do not

*Listing 1.* The frozen part of the evaluation prompt, which contains the expert demonstration episodes and stays constant throughout an evaluation episode. In this example, we have 8 demonstration episodes with 10 steps each and RGB observations. Before feeding the prompt to the model, we replace the observation and action placeholders with the actual observations (i.e., images in this case) and action strings.

```
1   demonstration_prompt = '''
2   You are a powerful reinforcement learning agent. You can effectively identify a policy
3   exposed by demonstrations and reproduce it in a new situation.
4
5   Here are a number of demonstrations:
6
7   Observation: <IMG_0_0> Action: <AC_0_0>
8   Observation: <IMG_0_1> Action: <AC_0_1>
9   ...
10  Observation: <IMG_0_9> Action: <AC_0_9>
11
12  Observation: <IMG_1_0> Action: <AC_1_0>
13  Observation: <IMG_1_1> Action: <AC_1_1>
14  ...
15  Observation: <IMG_1_9> Action: <AC_1_9>
16
17  ...
18
19  Observation: <IMG_7_0> Action: <AC_7_0>
20  Observation: <IMG_7_1> Action: <AC_7_1>
21  ...
22  Observation: <IMG_7_9> Action: <AC_7_9>
23  '''
```

*Listing 2.* The dynamic part of the evaluation prompt containing the evaluation trajectory. While stepping through an environment, we append this prompt to the one in Listing 1 in every evaluation step (e.g., for the 3rd step here), again replacing the observation and action placeholders with the actual observations and actions. This example also shows the legal moves (lines 8 to 10) and uses chain-of-thought prompting (lines 12 to 17, Wei et al. (2022)), both of which may be omitted depending on our ablations in Appendix C.4 (which can vary for each model-task combination).

```
1   evaluation_prompt = '''
2   This is the current situation:
3
4   Observation: <IMG_8_0> Action: <AC_8_0>
5   Observation: <IMG_8_1> Action: <AC_8_1>
6   Observation: <IMG_8_2>
7
8   In this situation, this is the list of all the moves that are legal:
9
10  no action, jump left, left
11
12  Given the demonstrations and the current situation, you should infer the next logical
13  action. Check that the chosen action is in the set of legal moves. Think step by step
14  and very briefly explain your reasoning for choosing this  action. You must answer with
15  the reasoning followed by the action in the following format:
16  Reasoning: ...
17  Action: ...
18  '''
```

```
{
  'position': [-0.09944233298301697, ..., -0.032929372042417526],
  'velocity': [0.0642457902431488, ..., 0.1964229941368103],
}
```

(a) Observation

```
[-0.8040763139724731, 0.9528138637542725, -1.0, 0.7932683229446411, 1.0, 1.0]
```

(b) Action

*Figure A5.* A sample observation and action from the cheetah run task from the DM Control suite (Tassa et al., 2018). Note that the observation is actually presented to the model on a single line (we use the multi-line representation here for ease of visualization). Moreover, we only show the truncated position and velocity vectors (represented by the ellipsis). The full position vector contains 8 elements, and the full velocity vector contains 9 elements. All elements are `float64` converted to string.

produce an output (in which case the API returns an empty sample). Since we want to report each model's best performance on our benchmark, we therefore ablate the maximum number of sample tokens and choose the configuration that trades off good performance and low cost (since, after a certain point, more tokens generally do not improve performance but only increase the cost). The maximum sample length also has an impact on the other models, particularly when using chain-of-thought reasoning, but a relatively low sample length typically suffices for our tasks (unlike the o1 models which require large maximal sample length).

To that end, Fig. A7 shows our ablation of the maximum sample length for Claude 3.5 Sonnet, GPT-4o, o1-mini, o1-preview, Gemini 1.5 Flash, and Gemini 1.5 Pro on all three observation types (ASCII, coordinates and RGB) for the grid world navigation task with 1 demonstration episode. Fig. A7 shows the average score over 100 evaluation episodes for maximum sample lengths from 32 to 32768 tokens (Claude 3.5 Sonnet only supports 8192 output tokens and GPT-4o only supports 16384 output tokens). Unsurprisingly, the performance of Claude 3.5 Sonnet, GPT-4o, Gemini 1.5 Flash, and Gemini 1.5 Pro is largely unaffected by the number of sample tokens, i.e., even 32 tokens are sufficient to achieve their best performance (with the exception of Claude 3.5 Sonnet and GPT-4o on RGB observations, where they benefit from having 64 tokens). This is in stark contrast with o1-mini and o1-preview, which require between 4096 and 8192 output tokens to achieve their optimal performance (with a very steep degradation below 4096). To verify whether this sharp rise in performance is actually due to the model not generating a valid action, Fig. A7 also visualizes the average percentage of illegal actions per episode over the maximum sample length, which (inversely) correlates very strongly with performance for the o1 models. We therefore set the maximum sample length to 8192 for the o1 models as a tradeoff between cost and performance (the maximum for o1-preview is 32768, the maximum for o1-mini is 65536 (OpenAI, 2024d)). This should enable the o1 models to achieve their best performance on our benchmark — even though they do so at a much higher (computational) cost than the other models.

### C.2. Replaying a Demonstration Episode

In-context imitation learning requires several different skills, one of which is being able to locate and retrieve the relevant demonstration(s) from the context. We therefore conduct a "sanity check" experiment where we provide a single demonstration episode and "replay" the exact same episode for evaluation (akin to a multimodal sequence copying task). Thus, for every step, the model only has to find the correct location in the demonstration episode in its context and reproduce its corresponding action. Accordingly, we slightly change the evaluation setup since the next observation (both action and next state) in the evaluation trajectory is always determined by the demonstration episode and not by the action generated by the model (i.e., we perform teacher forcing rather than a dynamic evaluation). As a result, we are not interested in a model's return but in how often it manages to match the action from the demonstration episode. Like in our other experiments, we evaluate 100 different episodes. For the sake of simplicity, we do not use chain-of-thought prompting, do not show the legal actions in the prompt, and only consider a single demonstration episode. Other than that, we leave the prompt the same as in our default experimental setup (i.e., including the same preamble and separator).

Figs. A8 to A13 show the average action replay accuracy per evaluation step for all frontier models on all six environments (for all state representation formats separately). We observe that models are generally capable of replaying the demonstration episodes, with slightly lower performance deeper into the episode. The significant exception is o1-mini which struggles with the replay task in most settings. Similarly, Gemini 1.5 Flash fails to replay the actions for tic-tac-toe with RGB

```
---------------------------------------------------------------------------------------------------------
| wall  | wall  | wall  | wall  | wall  | wall  | wall  | wall   | wall  | wall  | wall  | wall  |
---------------------------------------------------------------------------------------------------------
| wall  | tile  | tile  | tile  | tile  | tile  | tile  | tile   | tile  | tile  | tile  | wall  |
---------------------------------------------------------------------------------------------------------
| wall  | tile  | tile  | tile  | tile  | tile  | tile  | player | tile  | tile  | tile  | wall  |
---------------------------------------------------------------------------------------------------------
| wall  | tile  | tile  | tile  | tile  | tile  | tile  | tile   | tile  | tile  | tile  | wall  |
---------------------------------------------------------------------------------------------------------
| wall  | tile  | tile  | tile  | tile  | tile  | tile  | tile   | tile  | tile  | tile  | wall  |
---------------------------------------------------------------------------------------------------------
| wall  | tile  | tile  | tile  | tile  | tile  | tile  | tile   | tile  | tile  | tile  | wall  |
---------------------------------------------------------------------------------------------------------
| wall  | tile  |target | tile  | tile  | tile  | tile  | tile   | tile  | tile  | tile  | wall  |
---------------------------------------------------------------------------------------------------------
| wall  | tile  | tile  | tile  | tile  | tile  | tile  | tile   | tile  | tile  | tile  | wall  |
---------------------------------------------------------------------------------------------------------
| wall  | tile  | tile  | tile  | tile  | tile  | tile  | tile   | tile  | tile  | tile  | wall  |
---------------------------------------------------------------------------------------------------------
| wall  | tile  | tile  | tile  | tile  | tile  | tile  | tile   | tile  | tile  | tile  | wall  |
---------------------------------------------------------------------------------------------------------
| wall  | tile  | tile  | tile  | tile  | tile  | tile  | tile   | tile  | tile  | tile  | wall  |
---------------------------------------------------------------------------------------------------------
| wall  | wall  | wall  | wall  | wall  | wall  | wall  | wall   | wall  | wall  | wall  | wall  |
---------------------------------------------------------------------------------------------------------
```

(a) ASCII

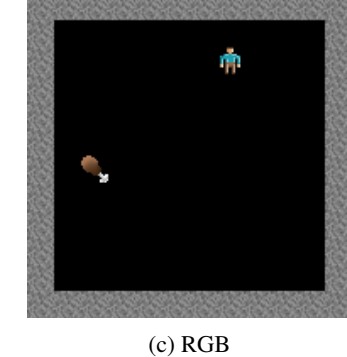

```
{'player': [2, 7], 'target': [6, 2]}
```

(b) Coordinates

(c) RGB

*Figure A6.* Sample observations from our grid world environment for all three state representation formats. For the RGB observations (Fig. A6c), we use the sprites from Crafter (Hafner, 2022) for the walls, the player, and the target (the floor is a black square).

observations. Note that, in theory, the observation type is irrelevant for this task since the models could just count the number of observations in the evaluation trajectory and select the corresponding action in the demonstration trajectory.

### C.3. Illegal Actions

As described in Section 2.4, we do not terminate the evaluation episode early if a model does not produce a valid action (except on the crossword task) but instead randomly sample one of the actions that are legal in the current state and continue the evaluation with the observation produced by that action. To differentiate illegal actions from acting randomly (over legal actions), we compute how often models actually propose an illegal action and visualize the percentage of illegal actions per episode over the number of expert demonstrations in Figs. A15 to A20 (analogous to Figs. 3 to 8). Note that, since we perform an ablation over *showing the legal moves in the prompt* (see Appendix C.4), in theory, all models could have the necessary information to sample a legal action at all times (whether we actually show this or not in our main experiments depends on whether it improves model performance in the ablations). Nevertheless, we observe that models do produce illegal actions actions across most environments. For example, on the grid world navigation task with coordinate observations o1-preview produces almost 100% illegal actions with 2 or more expert demonstrations in the context. Interestingly, the ability to produce legal actions also depends on the observation type: For tic-tac-toe, all models mostly produce legal actions with ASCII observations, but the Gemini 1.5 models generate up to $\sim$ 60% illegal actions for RGB observations from the same environment.

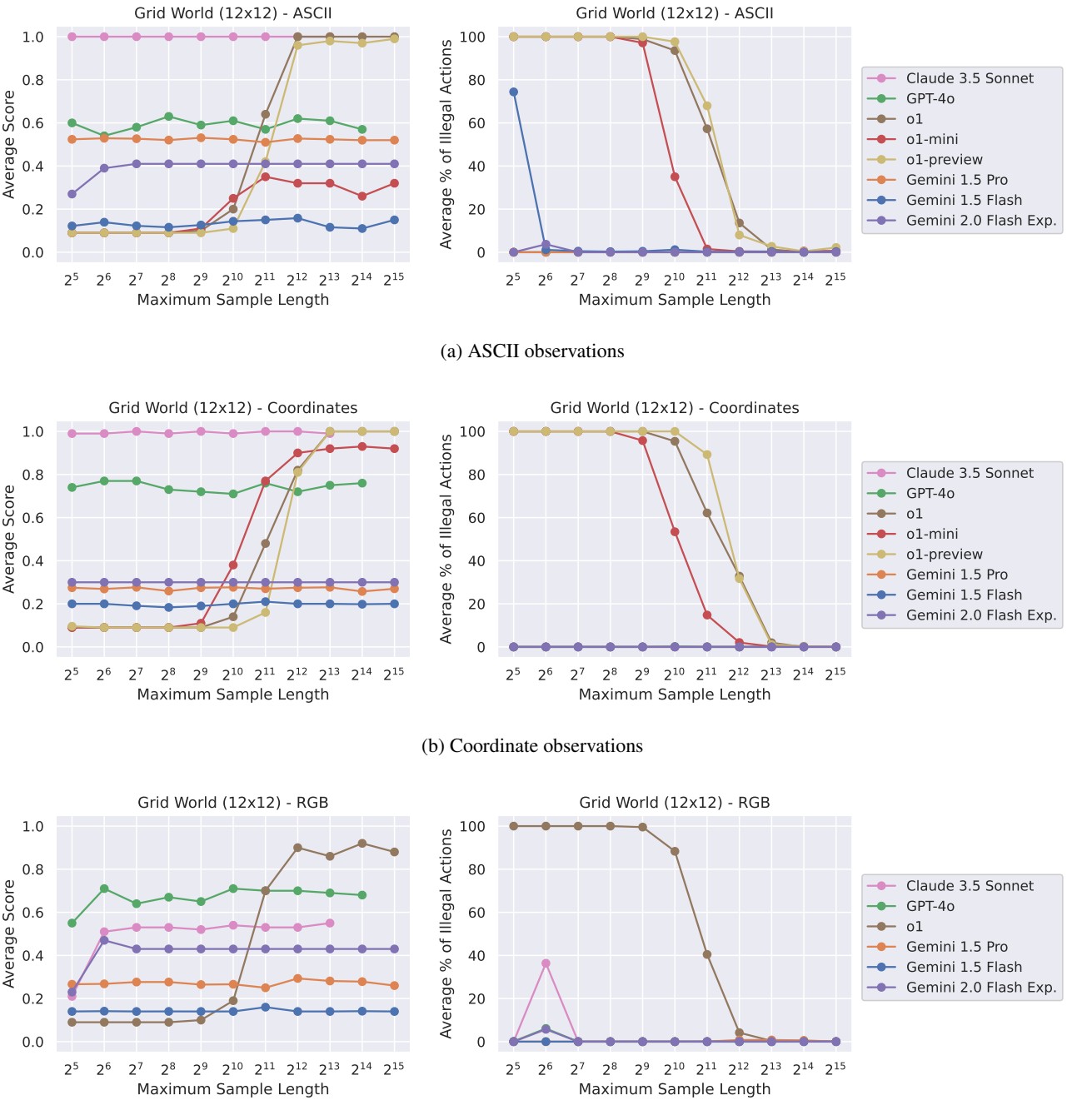

*Figure A7.* Ablating the maximum sample length for all three observation types (ASCII, coordinates, and RGB images) of the grid world navigation task with 1 demonstration episode (the left panels show the average score, the right panels show the percentage of illegal actions). As expected, 32 output tokens are sufficient for Claude 3.5 Sonnet, GPT-4o, Gemini 1.5 Flash, and Gemini 1.5 Pro to achieve their best performance on the task. In contrast, o1-mini and o1-preview require between 4096 and 8192 output tokens to achieve their best performance. With less than 8192 output tokens, the o1 models do not have enough internal "reasoning tokens" at their disposal to produce an output, which can be seen by the sharp increase in the percentage of illegal actions in the right panels. Note that o1-mini and o1-preview are text-only models and therefore cannot process RGB image observations.

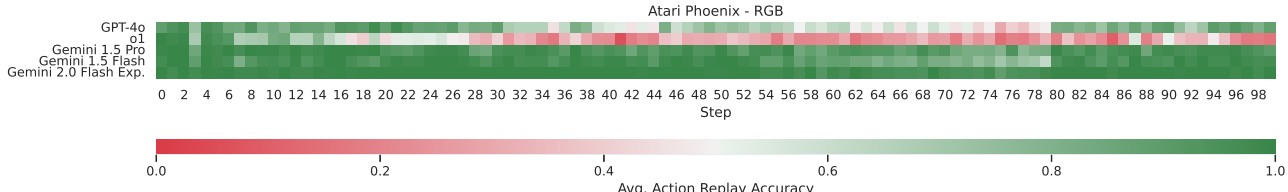

*Figure A8.* Replaying the demonstration episode for the Phoenix game (with RGB observations) from Atari 2600. The color visualizes the models' average accuracy when attempting to replay the action for a given step. The Gemini 1.5 models generally perform well at replaying the demonstrations, regardless of the step, though they show a slight degradation in performance towards the 80-step mark, from which they immediately recover again. The performance of GPT-4o shows a slightly higher degradation towards step 80, but also immediately recovers thereafter. Note that Claude 3.5 Sonnet cannot process more than 100 images at a time, so we cannot evaluate it on this task. Similarly, o1-mini and o1-preview are text-only and, therefore, cannot process RGB observations.

*Table A1.* Ablating the use of chain-of-thought prompting and whether or not to show legal actions in the prompt for the Phoenix game from Atari 2600. For this task, all models perform best without legal actions and without chain-of-thought.

| Model | Observation | Legal Actions | Chain-of-Thought | Average Score |
|---|---|---|---|---|
| Gemini 1.5 Flash | RGB | False | False | **42.83** |
| | | | True | 6.16 |
| | | True | False | 8.07 |
| | | | True | 6.46 |
| Gemini 1.5 Pro | RGB | False | False | **21.86** |
| | | | True | 8.68 |
| | | True | False | 6.33 |
| | | | True | 6.21 |
| Gemini 2.0 Flash Exp. | RGB | False | False | **31.80** |
| | | | True | 11.40 |
| | | True | False | 15.20 |
| | | | True | 10.20 |
| GPT-4o | RGB | False | False | **9.95** |
| | | | True | 5.40 |
| | | True | False | 6.40 |
| | | | True | 5.40 |
| o1 | RGB | False | False | **29.00** |
| | | | True | 19.20 |
| | | True | False | 23.00 |
| | | | True | 16.40 |

## C.4. Hyperparameter Ablations

As mentioned in Section 3, we ablate the use of chain-of-though prompting (Wei et al., 2022) and whether or not to include the list of actions that is legal given the current observation in the prompt for each model-task combination since we want to report the best-possible performance the models can attain on our benchmark. For completeness, we present all the ablation results in Tables A1 to A10. We highlight the best score for each model-observation pair in bold and use the corresponding hyperparameter combination for our sweeps over the number of expert demonstration episodes in Section 3.2.

### C.4.1. INCLUDING PAST ACTIONS

Over the course of our experimental investigation of frontier models' performance in dynamic environments, we noticed that these models have a tendency of getting stuck into repeating their previous action. We therefore conduct an ablation where we do not show the past actions of the evaluation trajectory in the prompt (the past observations remain in the prompt, as do the observations and actions of the demonstration episode). Tables A11 to A13, A15 and A16 show the performance for all frontier models on the grid world navigation task with 1 expert demonstration episode with and without the past actions in the prompt. In general, all models benefit from having the full history in the prompt (i.e., both the previous observations and actions), even if that allows them to repeat their previous action more easily (in most environments they can still infer the previous action from the two past observations). The only exception is Claude 3.5 Sonnet (see Table A11), which does not show a clear trend in terms of including the past actions. Given these results, we decide to always include the past actions in the prompt for all models and environments in all our other experiments.

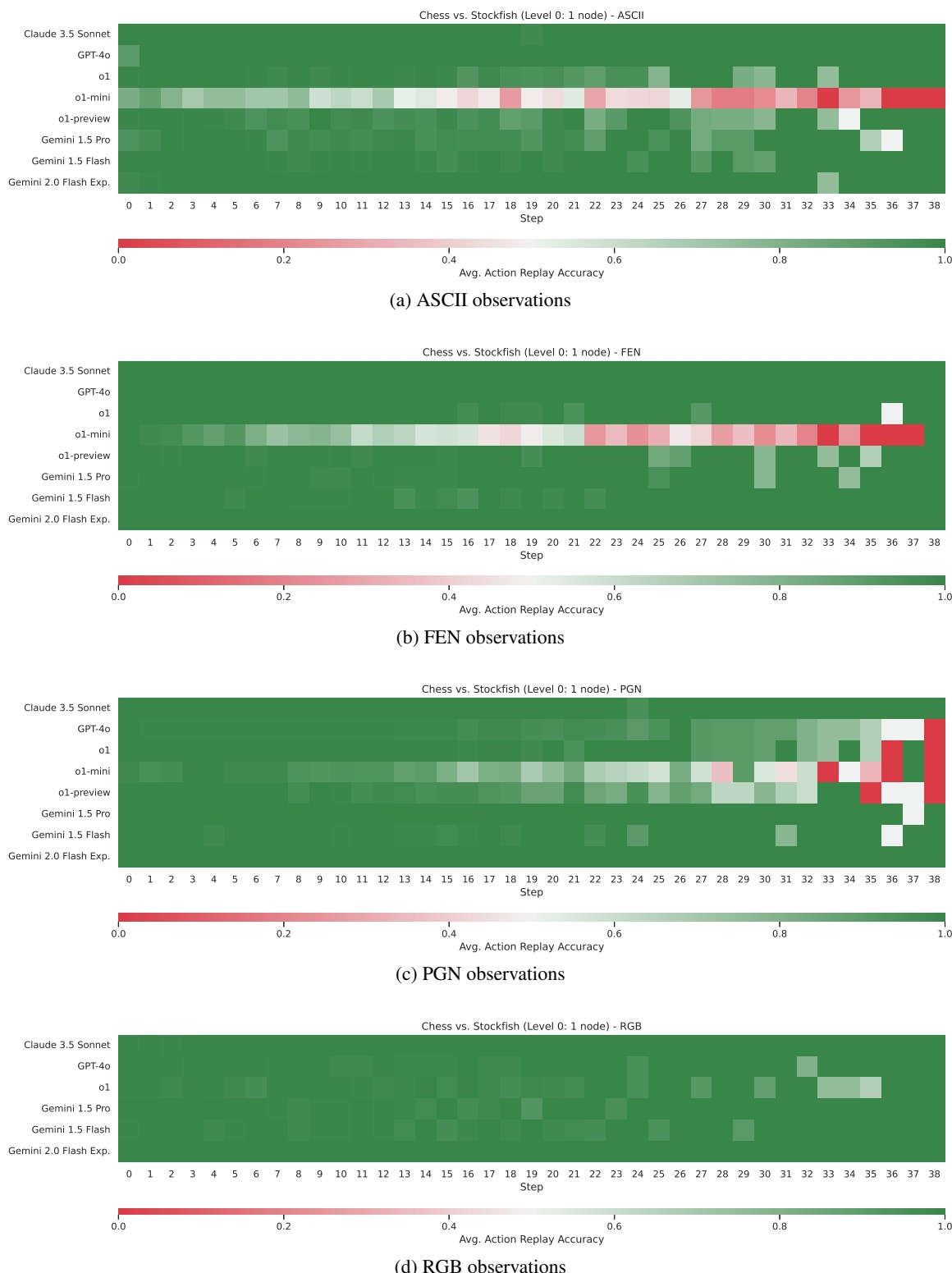

*Figure A9.* Replaying the demonstration episode for the different observation types from our chess environments. The color visualizes the models' average accuracy when attempting to replay the action for a given step. All models generally perform well across all observation types, except for o1-mini, which shows a strong performance degradation towards the end of the episode across all three observation text observation types (recall that o1-mini and o1-preview are text-only models, and, therefore, cannot process RGB images). With the exception of Claude 3.5 Sonnet, all models struggle (to varying degrees) to replay the last few actions with PGN observations.

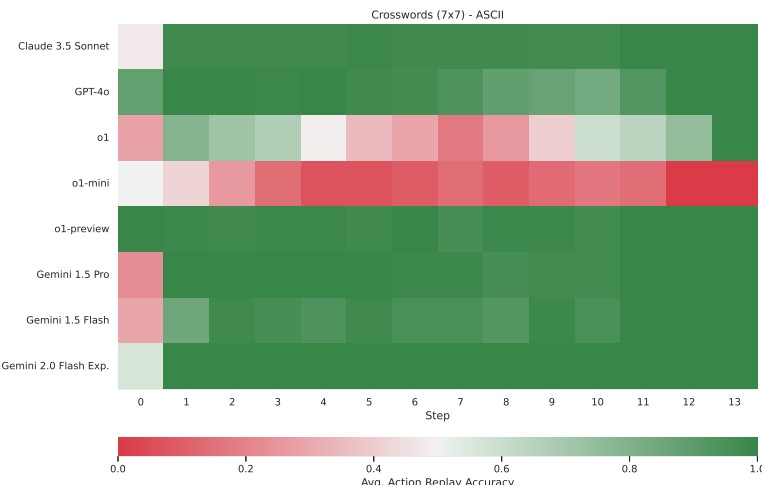

*Figure A10.* Replaying the demonstration episode for our crossword environment with ASCII observations. The color visualizes the models' average accuracy when attempting to replay the action for a given step. All models generally perform well, except for o1-mini, which completely fails at this task. Moreover, Claude 3.5 Sonnet, Gemini 1.5 Flash, and Gemini 1.5 Pro struggle to replay the first step.

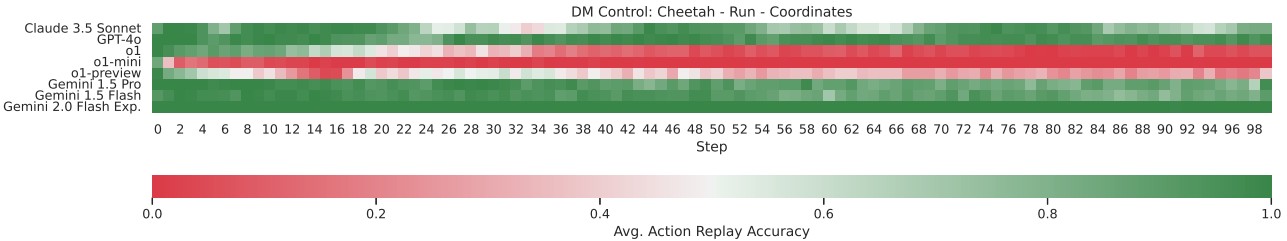

*Figure A11.* Replaying the demonstration episode for the cheetah run task from the DM Control suite (with coordinate observations). The color visualizes the models' average accuracy when attempting to replay the action for a given step. GPT-4o, Gemini 1.5 Flash, and Gemini 1.5 Pro generally perform well. In contrast, o1-mini and o1-preview significantly struggle with this task. Claude 3.5 Sonnet shows patches of slightly poorer performance around steps 32, 64, and 96, but performs well otherwise.

*Table A2.* Ablating the use of chain-of-thought prompting and whether or not to show legal actions in the prompt for Claude 3.5 Sonnet on chess. For this task, Claude 3.5 Sonnet always profits from having the legal actions in the prompt. Chain-of-thought prompting improves performance on FEN and RGB observations, but not on ASCII and PGN observations.

| Model | Observation | Legal Actions | Chain-of-Thought | Average Score |
|---|---|---|---|---|
| Claude 3.5 Sonnet (2024-10-22) | ASCII | False | False | -0.98 |
| | | | True | -0.99 |
| | | True | False | **-0.88** |
| | | | True | -0.93 |
| | FEN | False | False | -0.96 |
| | | | True | -0.99 |
| | | True | False | -0.96 |
| | | | True | **-0.95** |
| | PGN | False | False | -0.92 |
| | | | True | -0.97 |
| | | True | False | **-0.82** |
| | | | True | -0.86 |
| | RGB | False | False | -0.96 |
| | | | True | -0.99 |
| | | True | False | -0.94 |
| | | | True | **-0.91** |

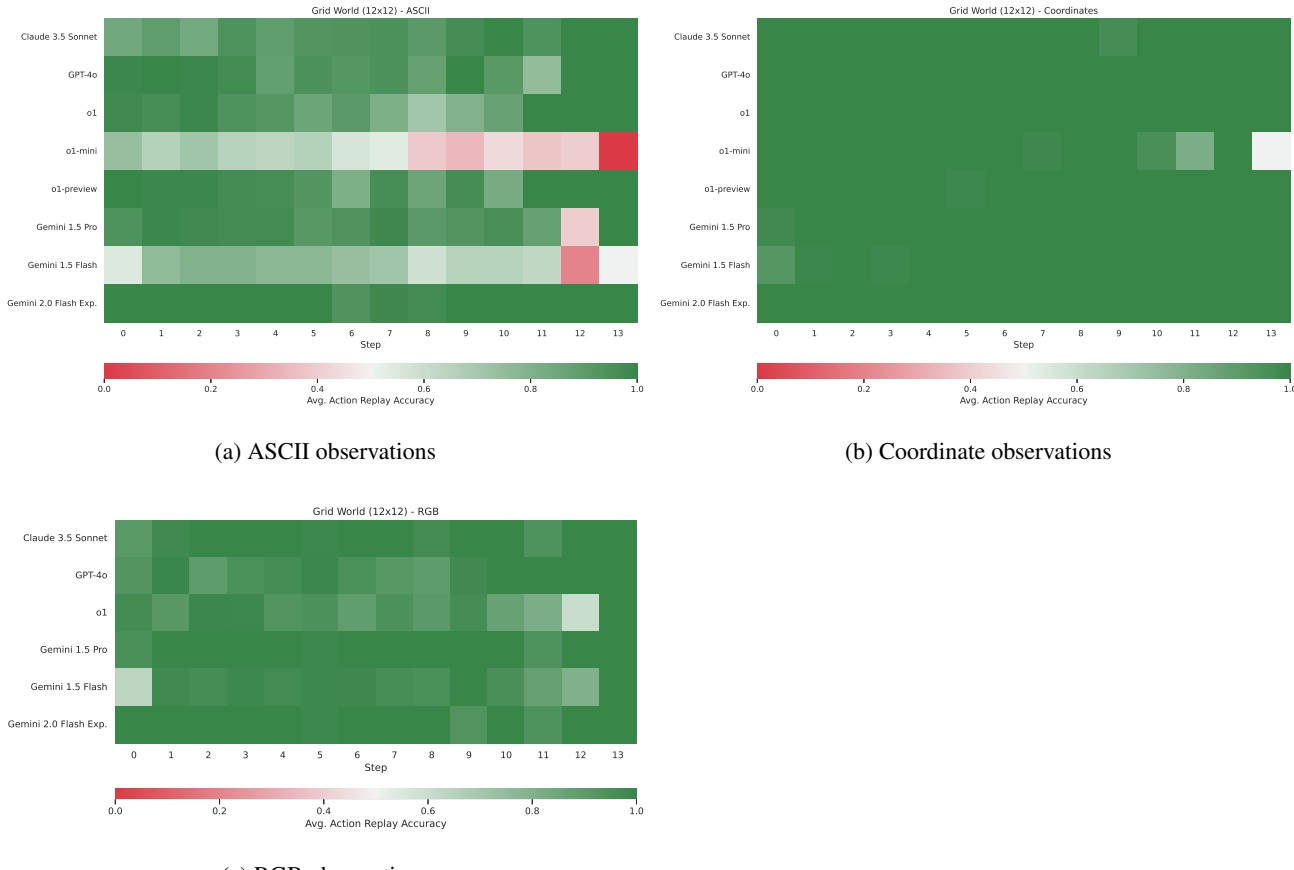

(a) ASCII observations

(b) Coordinate observations

(c) RGB observations

*Figure A12.* Replaying the demonstration episode for the all observation types from our grid world navigation task. The color visualizes the models' average accuracy when attempting to replay the action for a given step. All models generally perform well across all observation types, except for o1-mini on ASCII observations. Moreover, the Gemini 1.5 models struggle on step 12 with ASCII observations, and o1-mini struggles with the last step for coordinate observations. Note that o1-mini and o1-preview are text-only models and, therefore, cannot process RGB observations.

*Table A3.* Ablating the use of chain-of-thought prompting and whether or not to show legal actions in the prompt for Gemini 1.5 Flash, Gemini 1.5 Pro, and Gemini 2.0 Flash Experimental on chess. For this task, the models almost always profit from having the legal actions in the prompt. Chain-of-thought prompting sometimes improves performance, but not across the board.

| Model | Observation | Legal Actions | Chain-of-Thought | Average Score |
|---|---|---|---|---|
| Gemini 1.5 Flash | ASCII | False | False | -1.00 |
| | | | True | -1.00 |
| | | True | False | **-0.98** |
| | | | True | -0.99 |
| | FEN | False | False | -0.99 |
| | | | True | -0.99 |
| | | True | False | -0.99 |
| | | | True | **-0.98** |
| | PGN | False | False | -0.98 |
| | | | True | -0.99 |
| | | True | False | **-0.96** |
| | | | True | -0.97 |
| | RGB | False | False | -1.00 |
| | | | True | -1.00 |
| | | True | False | **-0.97** |
| | | | True | -1.00 |
| Gemini 1.5 Pro | ASCII | False | False | -1.00 |
| | | | True | -0.99 |
| | | True | False | **-0.96** |
| | | | True | -0.97 |
| | FEN | False | False | -1.00 |
| | | | True | -1.00 |
| | | True | False | **-0.95** |
| | | | True | -0.97 |
| | PGN | False | False | -0.93 |
| | | | True | -0.94 |
| | | True | False | **-0.90** |
| | | | True | -0.97 |
| | RGB | False | False | -1.00 |
| | | | True | -0.99 |
| | | True | False | -0.99 |
| | | | True | **-0.97** |
| Gemini 2.0 Flash Exp. | ASCII | False | False | -1.00 |
| | | | True | -0.99 |
| | | True | False | **-0.93** |
| | | | True | -0.98 |
| | FEN | False | False | -0.95 |
| | | | True | -0.99 |
| | | True | False | **-0.91** |
| | | | True | -0.97 |
| | PGN | False | False | **-0.87** |
| | | | True | -0.96 |
| | | True | False | -0.91 |
| | | | True | -0.96 |
| | RGB | False | False | -0.98 |
| | | | True | -0.99 |
| | | True | False | **-0.95** |
| | | | True | **-0.95** |

*Table A4.* Ablating the use of chain-of-thought prompting and whether or not to show legal actions in the prompt for GPT-4o, o1-mini, o1-preview, and o1 on chess. For this task, the models always profit from having the legal actions in the prompt. Chain-of-thought prompting sometimes improves performance, but not across the board.

| Model | Observation | Legal Actions | Chain-of-Thought | Average Score |
|---|---|---|---|---|
| GPT-4o | ASCII | False | False | -1.00 |
| | | | True | -1.00 |
| | | True | False | -0.94 |
| | | | True | **-0.92** |
| | FEN | False | False | -0.94 |
| | | | True | -0.98 |
| | | True | False | -0.98 |
| | | | True | **-0.90** |
| | PGN | False | False | **-0.63** |
| | | | True | -0.97 |
| | | True | False | -0.88 |
| | | | True | -0.93 |
| | RGB | False | False | -0.90 |
| | | | True | -0.99 |
| | | True | False | **-0.84** |
| | | | True | -0.91 |
| o1-mini | ASCII | False | False | -0.99 |
| | | | True | -0.99 |
| | | True | False | **-0.96** |
| | | | True | **-0.96** |
| | FEN | False | False | -1.00 |
| | | | True | -1.00 |
| | | True | False | **-0.89** |
| | | | True | -0.92 |
| | PGN | False | False | -0.99 |
| | | | True | -0.97 |
| | | True | False | -0.84 |
| | | | True | **-0.83** |
| o1-preview | ASCII | False | False | -0.99 |
| | | | True | -0.99 |
| | | True | False | **-0.93** |
| | | | True | -0.94 |
| | FEN | False | False | -0.99 |
| | | | True | -0.99 |
| | | True | False | -0.95 |
| | | | True | **-0.89** |
| | PGN | False | False | -0.98 |
| | | | True | -0.98 |
| | | True | False | **-0.95** |
| | | | True | -0.90 |
| o1 | ASCII | False | False | -0.98 |
| | | | True | -0.98 |
| | | True | False | **-0.85** |
| | | | True | -0.87 |
| | FEN | False | False | -0.95 |
| | | | True | -0.89 |
| | | True | False | **-0.82** |
| | | | True | -0.87 |
| | PGN | False | False | -0.80 |
| | | | True | -0.84 |
| | | True | False | -0.85 |
| | | | True | **-0.79** |
| | RGB | False | False | -0.95 |
| | | | True | -0.97 |
| | | True | False | **-0.82** |
| | | | True | -0.88 |

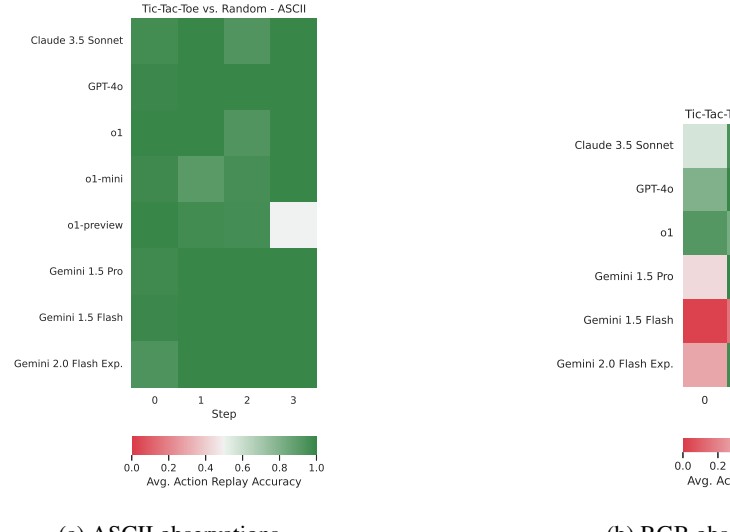

(a) ASCII observations                                    (b) RGB observations

*Figure A13.* Replaying the demonstration episode for the all observation types from our tic-tac-toe environment. The color visualizes the models' average accuracy when attempting to replay the action for a given step. All models generally perform well across all observation types except for Gemini 1.5 Flash on RGB images, where it fails completely. Claude 3.5 Sonnet and Gemini 1.5 Pro also struggle (to varying degrees) with the first step (step 0) for RGB observations. Note that o1-mini and o1-preview are text-only models and, therefore, cannot process RGB image observations.

*Table A5.* Ablating the use of chain-of-thought prompting for the task of solving crosswords. Chain-of-thought prompting sometimes improves performance, but not across all models. For this task, showing legal actions is infeasible (as it would include all possible (i.e., thousands of) words of the correct length for every slot), so we do not ablate it.

| Model | Observation | Legal Actions | Chain-of-Thought | Average Score |
|---|---|---|---|---|
| Claude 3.5 Sonnet (2024-10-22) | ASCII | False | False
True | 6.73
**6.81** |
| Gemini 1.5 Flash | ASCII | False | False
True | **1.99**
1.84 |
| Gemini 1.5 Pro | ASCII | False | False
True | 3.15
**5.50** |
| Gemini 2.0 Flash Exp. | ASCII | False | False
True | 3.70
**3.98** |
| GPT-4o | ASCII | False | False
True | 5.76
**6.67** |
| o1-mini | ASCII | False | False
True | **4.95**
4.82 |
| o1-preview | ASCII | False | False
True | 9.57
**9.79** |
| o1 | ASCII | False | False
True | 9.55
**9.56** |

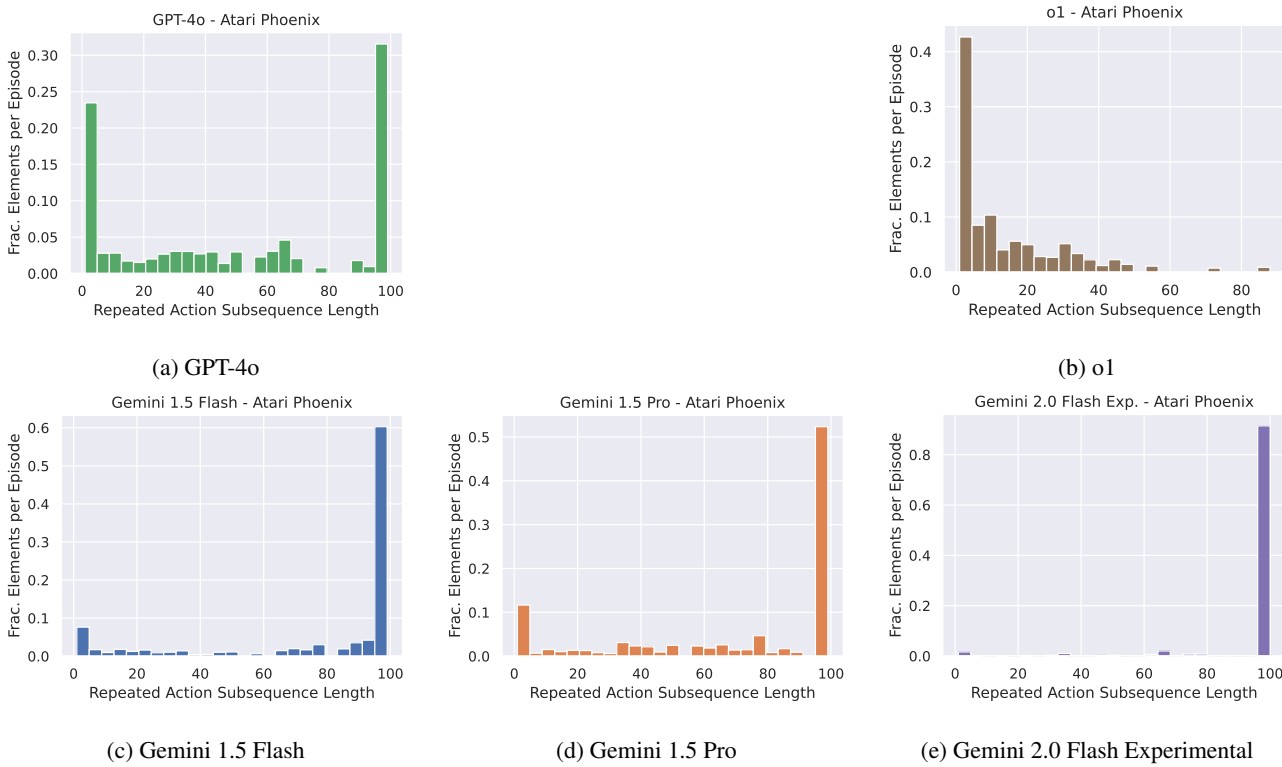

(a) GPT-4o

(b) o1

(c) Gemini 1.5 Flash

(d) Gemini 1.5 Pro

(e) Gemini 2.0 Flash Experimental

*Figure A14.* Fraction of elements per repeated action subsequence length in the Phoenix game from Atari 2600 (without demonstration episodes). To compute these results, we partition each evaluation episode into segments consisting of the same action generated consecutively. The segment length is on the x-axis, and the height of each bar is the fraction of actions (out of the total number of actions) across all time steps in all evaluation episodes that fall into each segment length. All three models have many episodes (e.g., more than 60% for Gemini 1.5 Flash) where they repeat the same action throughout almost the entire episode. Accordingly, they fire very rarely and thus achieve a low score (cf. Fig. 3), since, in the Phoenix game, in order to fire repeatedly, the firing button needs to be pressed and released repeatedly — constantly holding it down (i.e., repeating the previous action) only results in a single shot.

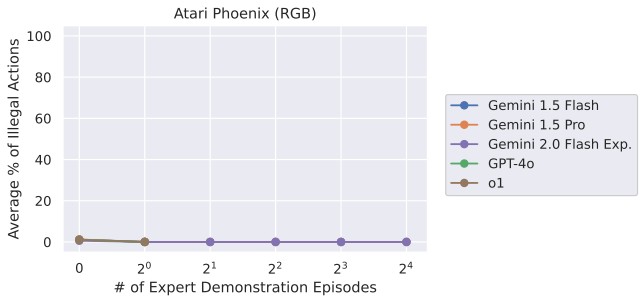

*Figure A15.* Average percentage of illegal actions per episode over the number of expert demonstrations for the Phoenix game (RGB observations) from Atari 2600. All models mostly generate legal actions. Note that o1-mini and o1-preview are text-only and, thus, cannot process RGB observations.

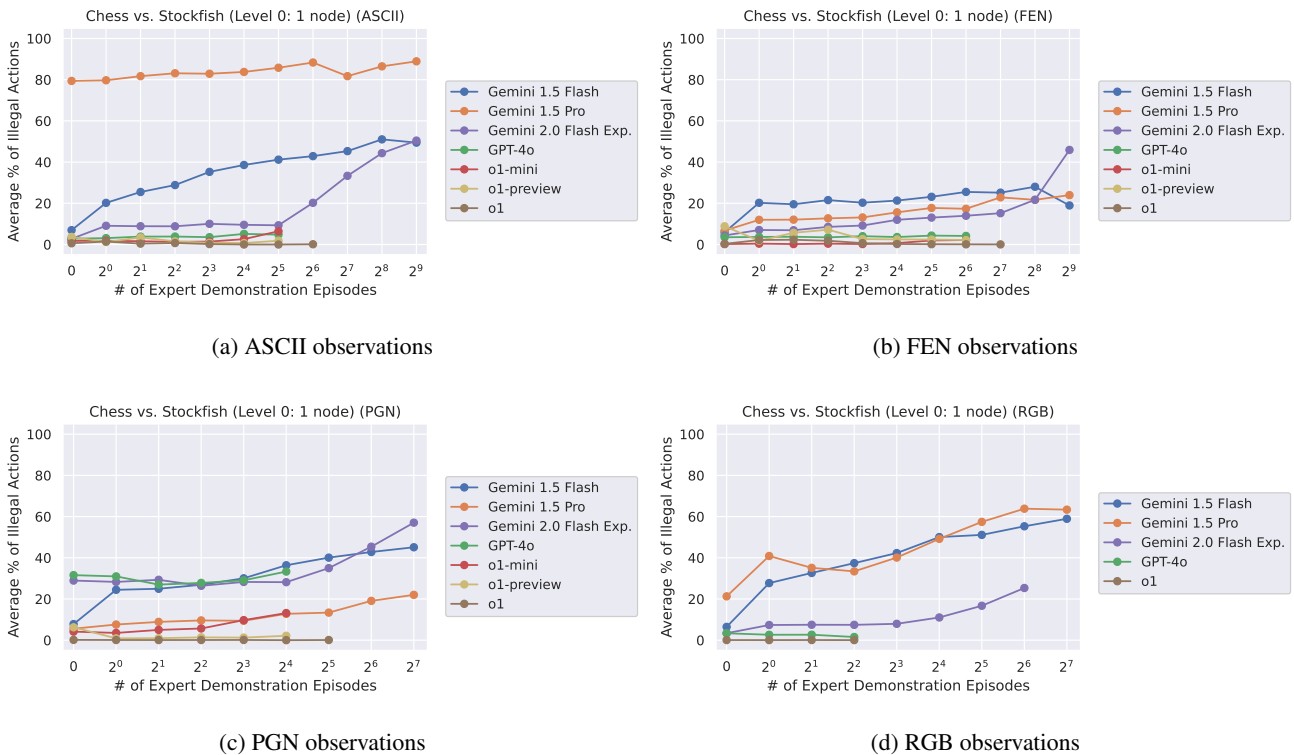

(a) ASCII observations

(b) FEN observations

(c) PGN observations

(d) RGB observations

*Figure A16.* Average percentage of illegal actions per episode over the number of expert demonstrations for all state representation formats from our chess environment. GPT-4o, o1-mini, and o1-preview rarely generate illegal actions (except for GPT-4o with PGN observations). In contrast, the Gemini 1.5 models consistently struggle to generate legal actions across all state representation formats and with higher rates of illegal actions with increasing numbers of expert demonstrations in the context. Note that o1-mini and o1-preview are text-only models and therefore cannot process RGB image observations.

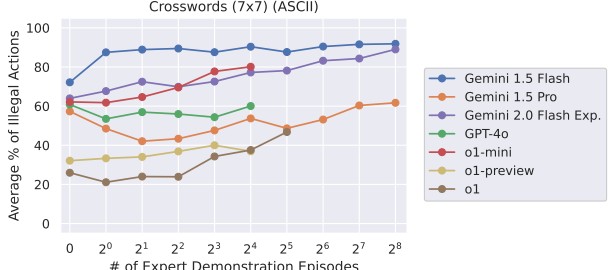

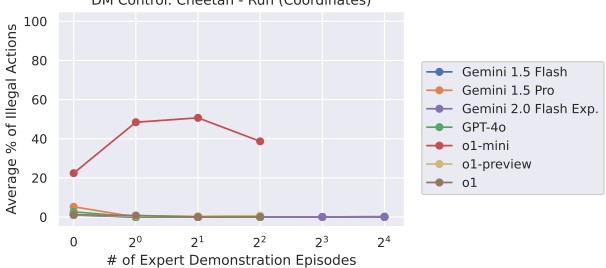

*Figure A17.* Average percentage of illegal actions per episode over the number of expert demonstrations for our crossword environment with ASCII observations. All models produce a high percentage of illegal actions (between ∼ 40% and ∼ 90% of all the actions in an episode).

*Figure A18.* Average percentage of illegal actions per episode over the number of expert demonstrations for the cheetah run task from the DM Control suite with coordinate observations. All models, except o1-mini, consistently produce mostly legal actions.

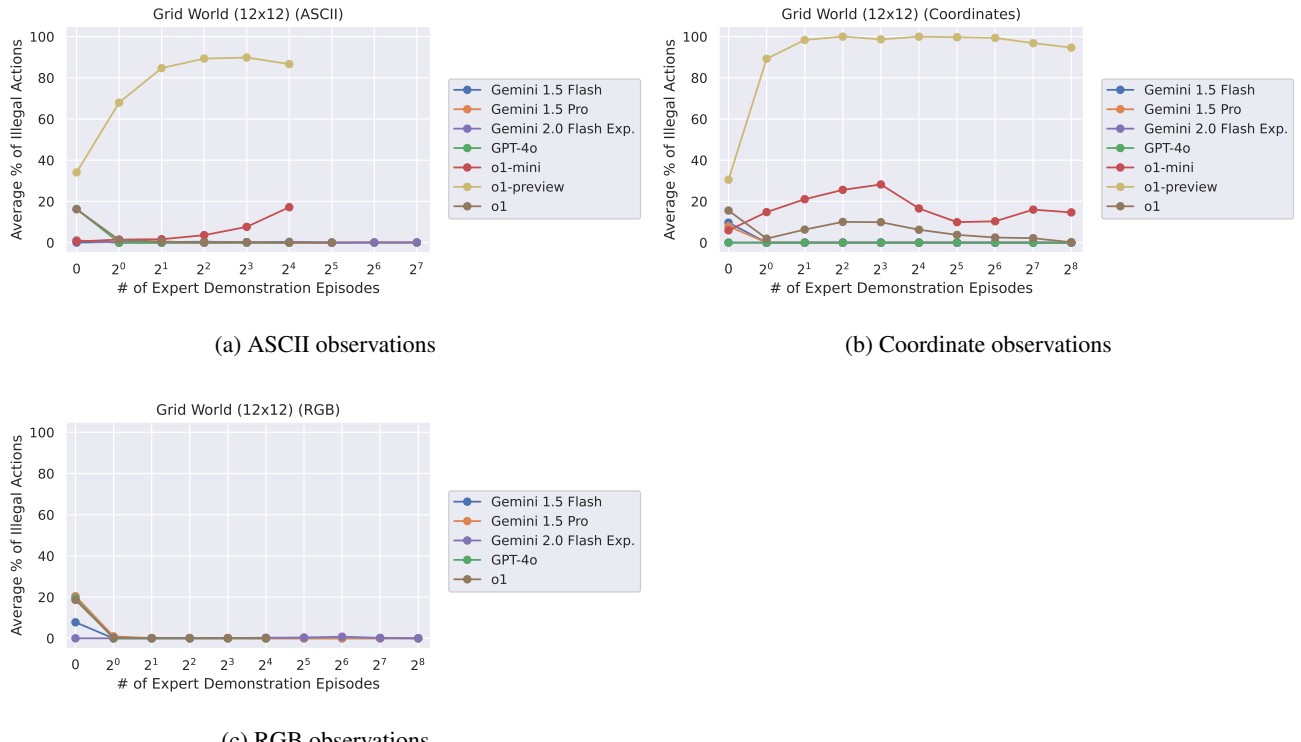

(a) ASCII observations

(b) Coordinate observations

(c) RGB observations

*Figure A19.* Average percentage of illegal actions per episode over the number of expert demonstrations for all state representation formats form our grid world navigation task. The o1 models increasingly struggle to produce legal actions with increasing numbers of expert demonstrations in the context, with o1-preview having a very high percentage of illegal actions. In contrast, GPT-4o, Gemini 1.5 Flash, and Gemini 1.5 Pro consistently produce legal actions. Note that o1-mini and o1-preview are text-only models and therefore cannot process RGB image observations.

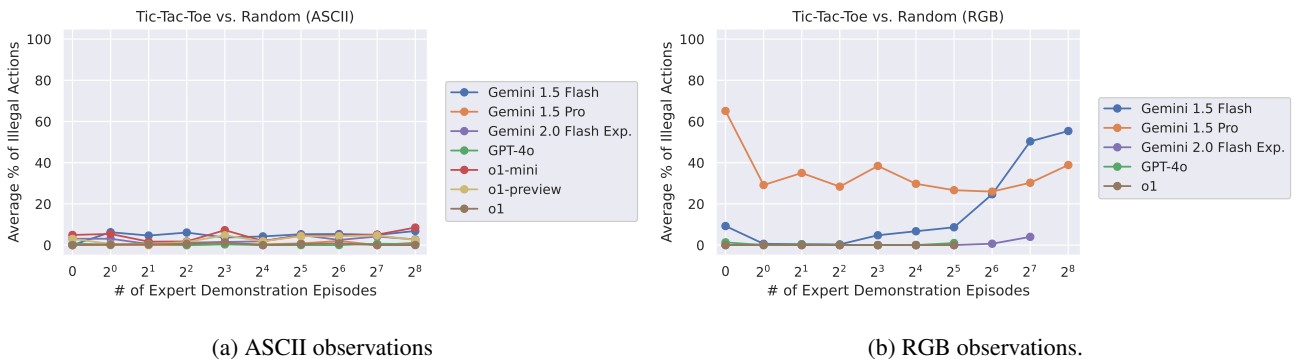

(a) ASCII observations

(b) RGB observations.

*Figure A20.* Average percentage of illegal actions per episode over the number of expert demonstrations for all state representation formats from our tic-tac-toe environment. With ASCII observations, all models mostly produce legal actions. In contrast, for RGB image observations, the Gemini 1.5 models struggle to produce legal actions (Gemini 1.5 Pro independently of the number of expert demonstrations and Gemini 1.5 Flash increasingly with the number of expert demonstrations). Note that o1-mini and o1-preview are text-only models and therefore cannot process RGB image observations.

*Table A6.* Ablating the use of chain-of-thought prompting and whether or not to show legal actions in the prompt for the task of simulating a cheetah from the DM Control Suite. For this task, most models do not benefit from chain-of-thought prompting or showing the legal actions in the prompt (note that there are infinitely many legal actions which we represent with the following string: `'A comma-separated list (enclosed by square brackets) of 6 values between -1 and 1.'`).

| Model | Observation | Legal Actions | Chain-of-Thought | Average Score |
|---|---|---|---|---|
| Claude 3.5 Sonnet (2024-10-22) | Coordinates | False | False | 4.70 |
| | | | True | 3.53 |
| | | True | False | **10.20** |
| | | | True | 1.33 |
| Gemini 1.5 Flash | Coordinates | False | False | **3.93** |
| | | | True | 1.89 |
| | | True | False | 0.39 |
| | | | True | 2.31 |
| Gemini 1.5 Pro | Coordinates | False | False | **18.00** |
| | | | True | 1.26 |
| | | True | False | 7.69 |
| | | | True | 2.44 |
| Gemini 2.0 Flash Exp. | Coordinates | False | False | **15.96** |
| | | | True | 1.91 |
| | | True | False | 13.53 |
| | | | True | 1.85 |
| GPT-4o | Coordinates | False | False | **11.03** |
| | | | True | 0.72 |
| | | True | False | 1.00 |
| | | | True | 0.92 |
| o1-mini | Coordinates | False | False | **0.80** |
| | | | True | 0.46 |
| | | True | False | 0.78 |
| | | | True | 0.44 |
| o1-preview | Coordinates | False | False | 3.16 |
| | | | True | 1.04 |
| | | True | False | **4.08** |
| | | | True | 1.88 |
| o1 | Coordinates | False | False | 2.08 |
| | | | True | **2.14** |
| | | True | False | 2.06 |
| | | | True | 1.84 |

*Table A7.* Ablating the use of chain-of-thought prompting and whether or not to show legal actions in the prompt for Claude 3.5 Sonnet on the grid world navigation task. For ASCII and coordinate observations, the performance is largely independent of the ablated settings. For RGB images, showing the legal actions and using chain-of-thought prompting performs best.

| Model | Observation | Legal Actions | Chain-of-Thought | Average Score |
|---|---|---|---|---|
| Claude 3.5 Sonnet (2024-10-22) | ASCII | False | False | **1.00** |
| | | | True | **1.00** |
| | | True | False | **1.00** |
| | | | True | **1.00** |
| | Coordinates | False | False | 0.96 |
| | | | True | **1.00** |
| | | True | False | 0.99 |
| | | | True | 0.99 |
| | RGB | False | False | 0.19 |
| | | | True | 0.40 |
| | | True | False | 0.21 |
| | | | True | **0.53** |

*Table A8.* Ablating the use of chain-of-thought prompting and whether or not to show legal actions in the prompt for Gemini 1.5 Flash, Gemini 1.5 Pro, and Gemini 2.0 Flash Experimental on the grid world navigation task. The best-performing setting is model and observation dependent.

| Model | Observation | Legal Actions | Chain-of-Thought | Average Score |
|---|---|---|---|---|
| Gemini 1.5 Flash | ASCII | False | False | 0.09 |
| | | | True | 0.08 |
| | | True | False | 0.11 |
| | | | True | **0.15** |
| | Coordinates | False | False | **0.21** |
| | | | True | 0.10 |
| | | True | False | 0.14 |
| | | | True | 0.08 |
| | RGB | False | False | **0.16** |
| | | | True | 0.12 |
| | | True | False | 0.14 |
| | | | True | 0.12 |
| Gemini 1.5 Pro | ASCII | False | False | **0.51** |
| | | | True | 0.25 |
| | | True | False | **0.51** |
| | | | True | 0.36 |
| | Coordinates | False | False | **0.27** |
| | | | True | 0.13 |
| | | True | False | 0.16 |
| | | | True | 0.12 |
| | RGB | False | False | 0.17 |
| | | | True | **0.25** |
| | | True | False | 0.24 |
| | | | True | 0.24 |
| Gemini 2.0 Flash Exp. | ASCII | False | False | 0.27 |
| | | | True | **0.41** |
| | | True | False | 0.25 |
| | | | True | 0.39 |
| | Coordinates | False | False | 0.22 |
| | | | True | 0.22 |
| | | True | False | **0.30** |
| | | | True | 0.16 |
| | RGB | False | False | 0.18 |
| | | | True | 0.37 |
| | | True | False | 0.23 |
| | | | True | **0.43** |

*Table A9.* Ablating the use of chain-of-thought prompting and whether or not to show legal actions in the prompt for GPT-4o, o1-mini, o1-preview, and o1 on the grid world navigation task. The best-performing setting is model and observation dependent. Note that o1-mini and o1-preview are text-only and, therefore, cannot process RGB observations.

| Model | Observation | Legal Actions | Chain-of-Thought | Average Score |
|---|---|---|---|---|
| GPT-4o | ASCII | False | False | 0.56 |
| | | | True | **0.57** |
| | | True | False | 0.50 |
| | | | True | 0.49 |
| | Coordinates | False | False | 0.69 |
| | | | True | 0.73 |
| | | True | False | **0.76** |
| | | | True | 0.72 |
| | RGB | False | False | 0.52 |
| | | | True | **0.70** |
| | | True | False | 0.56 |
| | | | True | 0.61 |
| o1-mini | ASCII | False | False | 0.20 |
| | | | True | 0.22 |
| | | True | False | 0.21 |
| | | | True | **0.32** |
| | Coordinates | False | False | **0.92** |
| | | | True | 0.79 |
| | | True | False | 0.87 |
| | | | True | 0.80 |
| o1-preview | ASCII | False | False | 0.94 |
| | | | True | **0.98** |
| | | True | False | 0.97 |
| | | | True | 0.97 |
| | Coordinates | False | False | 0.99 |
| | | | True | **1.00** |
| | | True | False | 0.99 |
| | | | True | **1.00** |
| o1 | ASCII | False | False | **1.00** |
| | | | True | **1.00** |
| | | True | False | **1.00** |
| | | | True | **1.00** |
| | Coordinates | False | False | 0.99 |
| | | | True | **1.00** |
| | | True | False | **1.00** |
| | | | True | **1.00** |
| | RGB | False | False | 0.81 |
| | | | True | **0.86** |
| | | True | False | 0.84 |
| | | | True | 0.84 |

*Table A10.* Ablating the use of chain-of-thought prompting and whether or not to show legal actions in the prompt for the game of tic-tac-toe. Across almost all models and observations types, using chain-of-thought prompting and showing the legal actions achieves the highest performance – often by a large margin. Note that o1-mini and o1-preview are text-only and, therefore, cannot process RGB observations.

| Model | Observation | Legal Actions | Chain-of-Thought | Average Score |
|---|---|---|---|---|
| Claude 3.5 Sonnet (2024-10-22) | ASCII | False | False | 0.10 |
| | | | True | 0.19 |
| | | True | False | 0.14 |
| | | | True | **0.34** |
| | RGB | False | False | -0.04 |
| | | | True | 0.16 |
| | | True | False | **0.20** |
| | | | True | 0.19 |
| Gemini 1.5 Flash | ASCII | False | False | -0.12 |
| | | | True | -0.15 |
| | | True | False | -0.10 |
| | | | True | **0.00** |
| | RGB | False | False | 0.07 |
| | | | True | 0.07 |
| | | True | False | -0.05 |
| | | | True | **0.10** |
| Gemini 1.5 Pro | ASCII | False | False | -0.08 |
| | | | True | -0.07 |
| | | True | False | **-0.02** |
| | | | True | -0.05 |
| | RGB | False | False | **0.08** |
| | | | True | 0.06 |
| | | True | False | -0.06 |
| | | | True | 0.01 |
| Gemini 2.0 Flash Exp. | ASCII | False | False | 0.06 |
| | | | True | 0.05 |
| | | True | False | 0.10 |
| | | | True | **0.18** |
| | RGB | False | False | -0.07 |
| | | | True | **0.07** |
| | | True | False | 0.06 |
| | | | True | 0.04 |
| GPT-4o | ASCII | False | False | 0.01 |
| | | | True | 0.15 |
| | | True | False | 0.11 |
| | | | True | **0.20** |
| | RGB | False | False | 0.00 |
| | | | True | 0.10 |
| | | True | False | 0.15 |
| | | | True | **0.17** |
| o1-mini | ASCII | False | False | 0.25 |
| | | | True | 0.26 |
| | | True | False | 0.32 |
| | | | True | **0.45** |
| o1-preview | ASCII | False | False | 0.21 |
| | | | True | 0.24 |
| | | True | False | 0.35 |
| | | | True | **0.36** |
| o1 | ASCII | False | False | 0.58 |
| | | | True | 0.58 |
| | | True | False | **0.83** |
| | | | True | 0.66 |
| | RGB | False | False | 0.38 |
| | | | True | 0.48 |
| | | True | False | 0.52 |
| | | | True | **0.60** |

*Table A11.* Ablating whether to include the past actions of the evaluation trajectory in the prompt (the actions of the demonstration episodes are always included) for Claude 3.5 Sonnet on our grid world navigation task. Models are prone to repeating the previous action, so omitting it from the prompt could alleviate this problem. For ASCII observations, Claude 3.5 Sonnet is mostly indifferent to having the actions in the prompt. For coordinate observations, Claude 3.5 Sonnet performs better with the actions in the prompt, while for RGB observations, the past actions deteriorate performance.

| Model | Observation | Legal Actions | Chain-of-Thought | Past Actions | Average Score |
|---|---|---|---|---|---|
| Claude 3.5 Sonnet (2024-10-22) | ASCII | False | False | False | 0.94 |
| | | | | **True** | **1.00** |
| | | | True | **False** | **1.00** |
| | | | | **True** | **1.00** |
| | | True | False | **False** | **1.00** |
| | | | | **True** | **1.00** |
| | | | True | **False** | **1.00** |
| | | | | **True** | **1.00** |
| | Coordinates | False | False | False | 0.77 |
| | | | | **True** | **0.96** |
| | | | True | False | 0.84 |
| | | | | **True** | **1.00** |
| | | True | False | False | 0.77 |
| | | | | **True** | **0.99** |
| | | | True | False | 0.78 |
| | | | | **True** | **0.99** |
| | RGB | False | False | **False** | **0.26** |
| | | | | True | 0.19 |
| | | | True | **False** | **0.47** |
| | | | | True | 0.40 |
| | | True | False | **False** | **0.37** |
| | | | | True | 0.21 |
| | | | True | **False** | **0.68** |
| | | | | True | 0.53 |

*Table A12.* Ablating whether to include the past actions of the evaluation trajectory in the prompt (the actions of the demonstration episodes are always included) for Gemini 1.5 Flash on our grid world navigation task. Models are prone to repeating the previous action, so omitting it from the prompt could alleviate this problem. In this case, including the full history of the current trajectory, i.e., both the observations and the actions, almost always improves the performance of Gemini 1.5 Flash.

| Model | Observation | Legal Actions | Chain-of-Thought | Past Actions | Average Score |
|---|---|---|---|---|---|
| Gemini 1.5 Flash | ASCII | False | False | False | 0.07 |
| | | | | **True** | **0.09** |
| | | | True | **False** | **0.12** |
| | | | | True | 0.08 |
| | | True | False | False | 0.09 |
| | | | | **True** | **0.11** |
| | | | True | False | 0.10 |
| | | | | **True** | **0.15** |
| | Coordinates | False | False | False | 0.08 |
| | | | | **True** | **0.21** |
| | | | True | False | 0.07 |
| | | | | **True** | **0.10** |
| | | True | False | False | 0.10 |
| | | | | **True** | **0.14** |
| | | | True | **False** | **0.08** |
| | | | | **True** | **0.08** |
| | RGB | False | False | False | 0.11 |
| | | | | **True** | **0.16** |
| | | | True | False | 0.08 |
| | | | | **True** | **0.12** |
| | | True | False | False | 0.10 |
| | | | | **True** | **0.14** |
| | | | True | False | 0.09 |
| | | | | **True** | **0.12** |

*Table A13.* Ablating whether to include the past actions of the evaluation trajectory in the prompt (the actions of the demonstration episodes are always included) for Gemini 1.5 Pro on our grid world navigation task. Models are prone to repeating the previous action, so omitting it from the prompt could alleviate this problem. Without exception, including the past actions in the prompt always improves the performance of Gemini 1.5 Pro.

| Model | Observation | Legal Actions | Chain-of-Thought | Past Actions | Average Score |
|-------|-------------|---------------|------------------|--------------|---------------|
| Gemini 1.5 Pro | ASCII | False | False | False
**True** | 0.41
**0.51** |
| | | | True | False
**True** | 0.22
**0.25** |
| | | True | False | False
**True** | 0.34
**0.51** |
| | | | True | False
**True** | 0.30
**0.36** |
| | Coordinates | False | False | False
**True** | 0.07
**0.27** |
| | | | True | False
**True** | 0.06
**0.13** |
| | | True | False | False
**True** | 0.02
**0.16** |
| | | | True | False
**True** | 0.08
**0.12** |
| | RGB | False | False | False
**True** | 0.14
**0.17** |
| | | | True | False
**True** | 0.14
**0.25** |
| | | True | False | False
**True** | 0.16
**0.24** |
| | | | True | False
**True** | 0.15
**0.24** |

*Table A14.* Ablating whether to include the past actions of the evaluation trajectory in the prompt (the actions of the demonstration episodes are always included) for Gemini 2.0 Flash Experimental on our grid world navigation task. Models are prone to repeating the previous action, so omitting it from the prompt could alleviate this problem. In this case, including the full history of the current trajectory, i.e., both the observations and the actions, almost always improves the performance of Gemini 2.0 Flash Experimental.

| Model | Observation | Legal Actions | Chain-of-Thought | Past Actions | Average Score |
|---|---|---|---|---|---|
| Gemini 2.0 Flash Exp. | ASCII | False | False | False | 0.19 |
| | | | | **True** | **0.27** |
| | | | True | False | 0.37 |
| | | | | **True** | **0.41** |
| | | True | False | False | 0.17 |
| | | | | **True** | **0.25** |
| | | | True | False | 0.36 |
| | | | | **True** | **0.39** |
| | Coordinates | False | False | False | 0.07 |
| | | | | **True** | **0.22** |
| | | | True | False | 0.08 |
| | | | | **True** | **0.22** |
| | | True | False | False | 0.06 |
| | | | | **True** | **0.30** |
| | | | True | False | 0.05 |
| | | | | **True** | **0.16** |
| | RGB | False | False | **False** | **0.25** |
| | | | | True | 0.18 |
| | | | True | False | 0.36 |
| | | | | **True** | **0.37** |
| | | True | False | **False** | **0.30** |
| | | | | True | 0.23 |
| | | | True | False | 0.38 |
| | | | | **True** | **0.43** |

*Table A15.* Ablating whether to include the past actions of the evaluation trajectory in the prompt (the actions of the demonstration episodes are always included) for GPT-4o on our grid world navigation task. Models are prone to repeating the previous action, so omitting it from the prompt could alleviate this problem. Including the past actions in the prompt always achieves the highest performance.

| Model | Observation | Legal Actions | Chain-of-Thought | Past Actions | Average Score |
|---|---|---|---|---|---|
| GPT-4o | ASCII | False | False | False | 0.28 |
| | | | | **True** | **0.56** |
| | | | True | False | 0.47 |
| | | | | **True** | **0.57** |
| | | True | False | False | 0.36 |
| | | | | **True** | **0.50** |
| | | | True | False | 0.43 |
| | | | | **True** | **0.49** |
| | Coordinates | False | False | False | 0.28 |
| | | | | **True** | **0.69** |
| | | | True | False | 0.46 |
| | | | | **True** | **0.73** |
| | | True | False | False | 0.21 |
| | | | | **True** | **0.76** |
| | | | True | False | 0.48 |
| | | | | **True** | **0.72** |
| | RGB | False | False | False | 0.47 |
| | | | | **True** | **0.52** |
| | | | True | False | 0.51 |
| | | | | **True** | **0.70** |
| | | True | False | False | 0.51 |
| | | | | **True** | **0.56** |
| | | | True | **False** | **0.61** |
| | | | | **True** | **0.61** |

*Table A16.* Ablating whether to include the past actions of the evaluation trajectory in the prompt (the actions of the demonstration episodes are always included) for o1-mini, o1-preview, and o1 on our grid world navigation task. Models are prone to repeating the previous action, so omitting it from the prompt could alleviate this problem. In the vast majority of cases, including the past actions in the evaluation prompt results in the best performance for o1-mini, o1-preview, and o1. Note that o1-mini and o1-preview are text-only models and, therefore, cannot process RGB images.

| Model | Observation | Legal Actions | Chain-of-Thought | Past Actions | Average Score |
|---|---|---|---|---|---|
| o1-mini | ASCII | False | False | False | 0.18 |
| | | | | **True** | **0.20** |
| | | | True | **False** | **0.22** |
| | | | | True | 0.15 |
| | | True | False | **False** | **0.24** |
| | | | | True | 0.23 |
| | | | True | False | 0.29 |
| | | | | **True** | **0.35** |
| | Coordinates | False | False | False | 0.41 |
| | | | | **True** | **0.68** |
| | | | True | False | 0.40 |
| | | | | **True** | **0.72** |
| | | True | False | False | 0.44 |
| | | | | **True** | **0.77** |
| | | | True | False | 0.44 |
| | | | | **True** | **0.71** |
| o1-preview | ASCII | False | False | False | 0.23 |
| | | | | **True** | **0.38** |
| | | | True | False | 0.31 |
| | | | | **True** | **0.42** |
| | | True | False | False | 0.30 |
| | | | | **True** | **0.42** |
| | | | True | False | 0.27 |
| | | | | **True** | **0.40** |
| | Coordinates | False | False | False | 0.10 |
| | | | | **True** | **0.13** |
| | | | True | **False** | **0.14** |
| | | | | True | 0.13 |
| | | True | False | False | 0.11 |
| | | | | **True** | **0.16** |
| | | | True | False | 0.10 |
| | | | | **True** | **0.13** |
| o1-mini | ASCII | False | False | **False** | **1.00** |
| | | | | **True** | **1.00** |
| | | | True | **False** | **1.00** |
| | | | | **True** | **1.00** |
| | | True | False | **False** | **1.00** |
| | | | | **True** | **1.00** |
| | | | True | **False** | **1.00** |
| | | | | **True** | **1.00** |
| | Coordinates | False | False | False | 0.98 |
| | | | | **True** | **0.99** |
| | | | True | False | 0.98 |
| | | | | **True** | **1.00** |
| | | True | False | False | 0.98 |
| | | | | **True** | **1.00** |
| | | | True | False | 0.99 |
| | | | | **True** | **1.00** |
| | RGB | False | False | False | 0.60 |
| | | | | **True** | **0.81** |
| | | | True | False | 0.59 |
| | | | | **True** | **0.86** |
| | | True | False | False | 0.65 |
| | | | | **True** | **0.84** |
| | | | True | False | 0.58 |
| | | | | **True** | **0.84** |

