# OpenReview forum: "LMAct: A Benchmark for In-Context Imitation Learning with Long Multimodal Demonstrations"
_ICML.cc/2025/Conference — ICML 2025 poster_

### Official Review · Reviewer_7Vdn · 2025-03-10

**Overall Recommendation:** 4

**Summary:**

LMAct is a benchmark designed to evaluate the multimodal in-context imitation learning capabilities of state-of-the-art, closed-source large multimodal foundation models (LMs). LMAct systematically evaluates model performance over extremely long-context inputs, testing how effectively these models utilize a varying number of expert demonstrations (spanning multiple orders of magnitude, up to context saturation). The benchmark includes six interactive decision-making tasks of varying complexity: Atari (Phoenix), Chess, Crosswords, DM Control (Cheetah Run), Grid World navigation, and Tic-Tac-Toe. Results suggest that the current state of the art multimodal LMs do struggle on the benchmark and in many cases, do not benefit from the additional context/demonstrations that are being provided.

**Claims And Evidence:**

Believe so.

**Essential References Not Discussed:**

The paper seems to overlook the exploration that researchers have done in the direction of in-context learning for control. State-based ICL has a long history (which has been applied to some of the environments listed in 2.2). See the line of work following One-Shot Imitation Learning [1] and Prompting Decision Transformer [2]. Multimodal (i.e. image-action-proprio) ICL for control using next-token prediction / autoregressive models have either a trained version (ICRT [3]) or are using pre-trained LLMs (Moka [4] prompt a robot to walk [6] and RoboPrompt [5]).

[1] Y. Duan et al., “One-shot imitation learning,” Advances in neural information processing systems, vol. 30, 2017.

[2] M. Xu et al., “Prompting decision transformer for few-shot policy generalization,” in international conference on machine learning, PMLR, 2022, pp. 24 631–24 645.

[3] F. Letian et al. "In-context imitation learning via next-token prediction." arXiv preprint arXiv:2408.15980 (2024).

[4] K. Fang, F. Liu, P. Abbeel, and S. Levine, “Moka: Open-world robotic manipulation through mark-based visual prompting,” Robotics: Science and Systems (RSS), 2024.

[5] Yin, Yida, Zekai Wang, Yuvan Sharma, Dantong Niu, Trevor Darrell, and Roei Herzig. "In-Context Learning Enables Robot Action Prediction in LLMs." arXiv preprint arXiv:2410.12782 (2024).

[6] Wang, Yen-Jen, Bike Zhang, Jianyu Chen, and Koushil Sreenath. "Prompt a robot to walk with large language models." In 2024 IEEE 63rd Conference on Decision and Control (CDC), pp. 1531-1538. IEEE, 2024.

**Experimental Designs Or Analyses:**

Since this is a benchmark paper, see the methods and evaluation criteria section above.

**Methods And Evaluation Criteria:**

This is a benchmark paper. The paper can benefit from more substantial compare and contrast with prior LLM benchmarks, especially those designed for multi-turn agents.

The paper also makes an effort to compare different multimodal representations of the state (i.e. ASCII and RGB observations). However,

The test contains 6 environments, which may not be broad enough and potentially can be saturated quickly. In addition, all environments are fully observable. It would be beneficial to 1) expand the number of environments and 2) introduce more partially observable environments.

**Other Comments Or Suggestions:**

N.A.

**Other Strengths And Weaknesses:**

N.A.

**Questions For Authors:**

Q1. I believe ICL for control is an exciting direction. Many prior works have worked towards using autoregressive models for ICL, including but not limited to the ones in the Essential References Not Discussed section. Adding appropriate discussion to these methods can help readers find better footing in the field.

Q2: The benchmark currently only has 6 environments. As mentioned earlier, the performance may saturate really quickly given the current climate of LLM research. Maybe consider expanding the benchmark or introducing harder tasks (i.e. partially observable instead of fully observable environments).

Q3: (Exploratory, less relevant to my evaluation of the paper) In all experiments, it seems that the temperature is set to zero except for the openai models. Does temperature affect ICL capability? Or does it just introduce higher variances?

**Relation To Broader Scientific Literature:**

I think this is particularly relevant to both the current trend of reasoning models and agents. It is great that this paper identifies areas/tasks where people can evaluate their reasoning models and language models on longer horizon tasks and seeing many of the current LMs not being able to in-context learn them allows the community to explore further in allowing better CoT + reasoning capabilities.

**Theoretical Claims:**

N.A.

---

> ### Author Rebuttal · Authors · 2025-04-01
>
> We thank the reviewer for their thorough assessment and insightful feedback. We are pleased that they think that our `paper is particularly relevant to the current trend of reasoning models` and that `it is great that this paper identifies areas where people can evaluate their reasoning models on longer horizon tasks`.
>
> **The paper overlooks the relevant line of work on in-context learning for control.**
>
> Indeed, the papers the reviewer listed are relevant to the broader context of our work, and we have expanded the related work section to discuss all of these references. While the methods explored in that exciting line of work focus on developing novel agents with strong in-context learning capabilities, our paper's primary goal is to benchmark the current state of existing, general-purpose frontier LMs on such tasks. Understanding how to best leverage insights from specialized agent research to enhance these large, pre-trained models remains an open question, which underscores the importance of establishing clear benchmarks like LMAct.
>
>
> **The paper only has 6 environments, which could lead to quick saturation given the current speed of AI research.**
>
> As discussed in Section 4, we designed our benchmark such that it can easily be made more difficult as the capabilities of frontier models evolve. While our benchmark is currently in its “easiest” form (e.g., grid world without obstacles, chess against Stockfish level 0 (less than 1300 Elo)), our results demonstrate that even this version reveals significant limitations in current frontier models. Therefore, we believe that our benchmark provides a strong and high-resolution signal to measure the progress of frontier models. Once our benchmark is saturated (which is far from being the case right now), we can easily construct a more challenging v2.
>
>
> **Consider including partially observable environments to expand the benchmark.**
>
> We tried to cover a range of environments that are relatively easy for humans but where current models still struggle to reach expert performance. As a result, we chose fully observable tasks (technically, Atari requires at least two frames to determine the velocities) since they have simple, reactive policies and do not require exploration.
>
> Partially observable tasks require consistent integration of information across several potentially non-adjacent time steps, which is a great test of a model’s ability to densely attend to the information in the context. Moreover, there is the theoretical problem of self-delusions in imitation of an expert that has hidden information (e.g., the hidden belief state of the agent), which is necessary under partial observability (see Ortega et al. (2021)). We want to keep this benchmark free from these complicating issues and think that the right time to move to such harder tasks is when frontier models easily solve simple, fully observable tasks at expert level (we are currently at beginner level — see the previous response). Independent of that, we agree with the reviewer that it is great to see benchmarks on interactive decision-making tasks under partial observability (perhaps better suited for in-context RL, which avoids the self-delusion problem) and refer to, e.g., the BALROG benchmark.
>
> We have added a brief discussion to acknowledge the value of partially observable benchmarks while clarifying our focus on fully observable tasks in the revised manuscript.
>
> *Shaking the foundations: delusions in sequence models for interaction and control.
> Pedro Ortega et al.
> https://arxiv.org/abs/2110.10819.*
>
>
> **Does the temperature affect the in-context learning capabilities?**
>
> We deliberately set the temperature to 0 to ensure the reproducibility of our results wherever possible (the API for the o1 models does not support this option). Whether the temperature also affects the performance is an interesting direction for future work. Note that since OpenAI’s models perform relatively well in most of our tasks, the effect of the temperature value does not seem to be very significant in our benchmark.

---

### Official Review · Reviewer_46g3 · 2025-03-10

**Overall Recommendation:** 4

**Summary:**

The paper presents a benchmark to evaluate the capabilities of today’s frontier models on multimodal decision making task in the very long context regime. The paper investigates the in-context learning abilities of these models. The authors compare a variety of the latest multimodal LM models on tasks like chess, crossword, and grid world, among others.

**Claims And Evidence:**

The paper investigates the performance of the latest LM models in very large context settings and studies the effect of the number of in-context demonstrations, different observation encoding schemes, and chain of thought prompting on the performance of these model. While the authors provide results for each, some insights about how to improve these models on the tasks being studied would be valuable for the community.

**Essential References Not Discussed:**

Being a research in a closely related but not the same field, the references seem adequate to me.

**Experimental Designs Or Analyses:**

Going through the paper at large, the experimental design seems sound. However, it would be great if the authors could include more analysis explaining the results and potentially provide pointers for improving the current models.

**Methods And Evaluation Criteria:**

Yes.

**Other Comments Or Suggestions:**

- From section 2.1, the o1 models use a larger context length (8192 tokens) than the other models (2048 tokens). Could this be the reason why they perform better than the other models on platforms like Crosswords and Tic-Tac-Toe?
- Are rewards from the demonstrations also given as input in the context? If yes, are they given as per-step rewards or as an accumulated value (similar to value functions in reinforcement learning)?
- Are 100 steps in an episode enough for most tasks? What about tasks that require more steps (for example in robotics where an episode could go on for a longer time)? Also, how did the authors come up with 100 steps? Can it be attributed to the limited context length of the current models?
- Can the authors provide an intuition of why there are a lot of illegal actions in chess and crosswords? Could adding the rules of the game in context be helpful in this case?
- It would be great if the authors could provide an insight into why the performance degrades with an increasing number of demonstrations in DMC cheetah run.

**Other Strengths And Weaknesses:**

Strengths
- The authors address an important problem of evaluating the latest frontier models on a series of long-context tasks.
- The paper covers a wide range of tasks including games (Atari, chess, tic-tac-toe), grid environments (grid world, tic-tac-toe, crosswords), and physical simulators (DMC cheetah run).
- The paper also considers a wide variety of frontier models for the comparisons, making for a comprehensive study.
- The paper also studies the effect of different observation encoding strategies, number of in-context demonstrations, and chain of thought prompting.

Weaknesses
- It would be great if the authors could include more analysis about failure models and potential improvement strategies to further advance the current state of these frontier models.

**Questions For Authors:**

It would be great if the authors could address the questions from the previous section and in Weaknesses. I would be happy to raise my score after these questions are addressed.

**Relation To Broader Scientific Literature:**

The results provided in the paper are valuable to the community for understanding the performance limitations of the latest frontier models. As mentioned above, more analysis of the results would be valuable to the community. For instance, since a lot of tasks have performances much lower than the expert performance, an analysis of common failure modes along with potential improvement strategies would be useful.

**Theoretical Claims:**

No theoretical claims.

---

> ### Author Rebuttal · Authors · 2025-04-01
>
> We thank the reviewer for their careful assessment and constructive feedback. We are pleased that they think our `paper addresses an important problem`, our `paper covers a wide range of tasks`, and our `paper considers a wide variety of frontier models, making for a comprehensive study`.
>
> **Could you include more analysis about failure modes and potential improvements?**
>
> We agree that one of the most important follow-ups is to analyze why models fail on our benchmarks and how to address these failures. We have already conducted a substantial set of such experiments in Appendix C, including:
> * Checking whether models can successfully “replay” (or copy) an episode, which most models can (Appendix C.2)
> * Checking to which degree bad performance can be attributed to the percentage of illegal vs. legal but suboptimal actions (Appendix C.3)
> * Investigating the fraction of repeated actions in Atari – Phoenix, which reveals that models have a high tendency to repeat the previous actions, which leads to few shots being fired, explaining why the performance is worse than the random action baseline (Figure A14)
> * Investigating whether failures can be attributed to various hyperparameter settings, e.g., checking whether showing legal actions alleviates the problem of having to infer the correct action format from the demonstrations (Appendix C.4)
>
> The trends of these investigations are complex (often model and task-dependent), and definitive answers would require sophisticated further analysis, which we consider outside the scope of our benchmark paper. We have added a compact discussion of this analysis to Section 4 in the updated manuscript.
>
> **Do the o1 models perform better because they use a larger context length (8192 tokens) than the other models (2048 tokens)?**
>
> Section 2.1 states that the o1 models use a larger *output sample length* (i.e., the maximum number of tokens to generate, including the “thinking tokens” for o1), not a larger context length. The (input) context length is the same across all models. We ablated the maximum sample length in Figure A7 and showed that only the o1 models benefit from a larger sample length. This is because the o1 models cannot finish their reasoning traces if they do not have sufficient “thinking tokens” at their disposal. Since our goal is to evaluate the models’ best possible performance, we set the output sample length to 8192 for the o1 models (which is expensive) and 2048 for all other models (since this is cheaper and does not affect their performance).
>
> **Do you also provide the rewards from the demonstrations?**
>
> No, we only perform in-context imitation learning, so we do not provide reward information to the models. We only use the rewards to compute the model scores for our benchmark results. We consider in-context reinforcement learning an interesting direction for future work.
>
> **Are the 100 steps per episode sufficient for most tasks? Can this number be attributed to the limited context length of current models?**
>
> Yes, for chess, crossword puzzles, gridworld navigation, and tic-tac-toe, 100 steps are more than sufficient to complete an episode (e.g., for chess, the average number of steps per game is 38 — see Section 2.2). However, for Atari and DM Control, we would ideally want to evaluate more steps (e.g., we only evaluate roughly 6 seconds of Atari play — see Section 2.2), but the context size limitations of current frontier models (32 Atari demonstrations episodes with 100 steps already hit the 1M token limit) do not allow this evaluation. We chose a maximum of 100 steps to be consistent across tasks while still providing a meaningful numerical signal (e.g., the models already fail in the early stages of Atari).
>
> **Why are there a lot of illegal actions in chess and crosswords? Could adding the rules of the game to the context be helpful?**
>
> The action space for these two tasks is more complex than for the other tasks (maybe except for Cheetah Run, which most models seem to be able to match, though). Adding game-specific explanations would probably help increase the performance, but our study is to investigate task-agnostic in-context learning behavior since such game-specific behavior may not always be available for real-world tasks (e.g., videos of human experts performing tasks in dynamic environments).
>
> **Why does the performance degrade with increasing demonstrations for Cheetah Run?**
>
> We do not know with certainty. Figure A18 shows that it cannot be attributed to an increased number of illegal actions, i.e., the models’ actions are increasingly suboptimal rather than illegal. A definitive answer to what causes in-context interference would require a thorough investigation, which is beyond the scope of this benchmark paper, but we have added a brief discussion of these observations and open questions to the paper.
>
> We hope our responses and revisions effectively address the reviewer’s concerns and clarify the paper's contribution.

---

### Official Review · Reviewer_YX5H · 2025-03-13

**Overall Recommendation:** 4

**Summary:**

The authors created a benchmark for empirical evaluation of the multimodal in-context imitation learning capabilities of some state-of-the-art LLMs (Claude 3.5 Sonnet, Gemini 1.5 Flash, Gemini 1.5 Pro, Gemini 2.0 Flash Experimental, GPT-4o, o1-mini, o1-preview, and o1) on several of interactive decision-making tasks: playing tic-tac-toe, chess, and Atari, navigating grid worlds, solving crosswords, and a DM Control task.
The authors show that even when optimizing the prompt (number of demonstrations, chain-of-thought prompting, etc.) for each model and task - frontier LLMs fail to reach expert performance on Atari, chess, and DM Control. Some models approach expert performance on crosswords, grid world, and tic-tac-toe. All models beat the random action baselines except on Atari.
The authors vary the number of expert demonstration episodes in the context from 0 up to 512 (the limit depends on the model and the task) and find that performance is mostly independent of the number of demonstrations. In some cases we observe strong in-context learning.
Authors run a control experiment where LLM agents need to replay the single demonstration episode in the context, where all models except for o1-mini perform well.
The authors plan to make their benchmark publicly available.

**Claims And Evidence:**

The authors created a benchmark that tested some existing Large Language Models, but the existing SOTA models also include QWEN [1], LLAMA [2].
The benchmark is aimed at studying models on a long context, but the authors did not compare their approach with another existing benchmark that allows studying large language models on long contexts – BabiLong [3].

[1] Bai, J., Bai, S., Chu, Y., Cui, Z., Dang, K., Deng, X., ... & Zhu, T. (2023). Qwen technical report. arXiv preprint arXiv:2309.16609
[2] Grattafiori, A., Dubey, A., Jauhri, A., Pandey, A., Kadian, A., Al-Dahle, A., ... & Vasic, P. (2024). The llama 3 herd of models. arXiv preprint arXiv:2407.21783.
[3] Kuratov, Y., Bulatov, A., Anokhin, P., Rodkin, I., Sorokin, D., Sorokin, A., & Burtsev, M. (2024). Babilong: Testing the limits of llms with long context reasoning-in-a-haystack. Advances in Neural Information Processing Systems, 37, 106519-106554.

**Essential References Not Discussed:**

The article lacks references to articles devoted to other SOTA LLMs: QWEN [1], LLAMA[2].
There is no reference to the benchmark for studying models on a long context Babilong [3].
There is no mention of specialized transformer models, specially developed for similar tasks for working on a long context: RMT [4], RATE [5].

[1] Bai, J., Bai, S., Chu, Y., Cui, Z., Dang, K., Deng, X., ... & Zhu, T. (2023). Qwen technical report. arXiv preprint arXiv:2309.16609
[2] Grattafiori, A., Dubey, A., Jauhri, A., Pandey, A., Kadian, A., Al-Dahle, A., ... & Vasic, P. (2024). The llama 3 herd of models. arXiv preprint arXiv:2407.21783.
[3] Kuratov, Y., Bulatov, A., Anokhin, P., Rodkin, I., Sorokin, D., Sorokin, A., & Burtsev, M. (2024). Babilong: Testing the limits of llms with long context reasoning-in-a-haystack. Advances in Neural Information Processing Systems, 37, 106519-106554.
[4] Bulatov, A., Kuratov, Y., & Burtsev, M. (2022). Recurrent memory transformer. Advances in Neural Information Processing Systems, 35, 11079-11091.
[5] Cherepanov, E., Staroverov, A., Yudin, D., Kovalev, A. K., & Panov, A. I. (2023). Recurrent action transformer with memory. arXiv preprint arXiv:2306.09459.]

**Experimental Designs Or Analyses:**

The soundness and validity of the experimental design are beyond doubt. At the same time, the lack of experiments with some other SOTA models (QWEN, LLAMA, etc.) is a drawback.

**Methods And Evaluation Criteria:**

The work is devoted to the creation of a large-scale benchmark, in general, the estimates that it allows to obtain are advective and can be applied to the assessment of large language models in the future. At the same time, the practical usefulness of this benchmark raises questions, because if the model successfully solves typical game problems given in the benchmark, then its portability for controlling complex real intelligent agents operating in a real environment (for example, robotic agents).

**Other Comments Or Suggestions:**

No other comments or suggestions.

**Other Strengths And Weaknesses:**

The originality of the work arises from creative combinations of existing environments for studying problems on a long context

The main weakness of the work is the insufficient completeness of comparison with existing LLMs, as well as the insufficient applicability to real-world problems.

**Questions For Authors:**

I would like to know from the authors whether it is possible to test transformer models created for solving LongContext tasks such as RMT[1], RATE[2] on their benchmark and what their quality indicators would be?

[1] Bulatov, A., Kuratov, Y., & Burtsev, M. (2022). Recurrent memory transformer. Advances in Neural Information Processing Systems, 35, 11079-11091.
[2] Cherepanov, E., Staroverov, A., Yudin, D., Kovalev, A. K., & Panov, A. I. (2023). Recurrent action transformer with memory. arXiv preprint arXiv:2306.09459.]

**Relation To Broader Scientific Literature:**

Key contributions of the paper are related to the broader scientific literature. The authors provided a fairly comprehensive analysis of existing approaches.

**Theoretical Claims:**

The work does not contain theoretical novelty, the main provisions of the work are related to obtaining empirical and practical results.

---

> ### Author Rebuttal · Authors · 2025-04-01
>
> We thank the reviewer for their thorough review, constructive feedback, and pointers to additional relevant related work. We are pleased that they think that `the soundness and validity of the experimental design are beyond doubt` and that `the originality of the work arises from creative combinations of existing environments to study problems in a long context`.
>
> **Why did you not evaluate the two other state-of-the-art models QWEN and Llama 3?**
>
> We had the dilemma of choosing a set of models to evaluate and ultimately settled on *8* models but did not mean to imply that any models outside of this set are not state-of-the-art. Unfortunately, we were unable to evaluate Llama 3 due to licensing issues, which is why we decided to focus on closed-weights models (which we explicitly stated in Section 2.1). Nevertheless, we fully agree that both QWEN and Llama 3 would be interesting additions to our benchmark and have discussed them in our revised manuscript. Since the landscape of state-of-the-art models is rapidly evolving, we open-sourced our benchmark to enable the community to evaluate QWEN, Llama 3, and any other future models using our approach.
>
>
> **Why did you not evaluate transformers created for long-context tasks (such as RMT and RATE) on your benchmark?**
>
> As the reviewer points out, RMT and RATE are specialized transformer models that were specifically developed for long-context tasks. However, as stated in Section 1, our goal is to evaluate *state-of-the-art LMs* in dynamic environments. Nevertheless, we agree that innovations around augmenting transformers with recurrency (and forms of more explicit retrieval) are very interesting approaches to potentially overcome LMs’ performance limitations in our benchmark, and we have expanded the related work section to discuss these approaches. Similar to QWEN and Llama 3, we are excited to see how the community will leverage the open-source nature of our benchmark to improve their models.
>
>
> **What is the practical usefulness of this benchmark?**
>
> We thank the reviewer for raising the important point regarding practical usefulness and transferability. While the LMAct environments are simpler than complex real-world scenarios like robotics, they require fundamental agentic capabilities (long-context multimodal understanding, in-context imitation learning) that are often tested in robotics research and likely necessary for real-world success. LMAct serves as a controlled testbed to evaluate these core skills, diagnose current model limitations, and guide research toward more generally capable agents. Although direct transfer is beyond this paper's scope, these findings can inform future work addressing that challenge. We have added a brief discussion clarifying this focus in the revised manuscript.
>
>
> **Why did you not compare your benchmark to BabiLong?**
>
> The BabiLong benchmark is relevant in the wider context of our work, and we have added a brief discussion in the related work section. Thanks for the pointer!

---

> > ### Comment · Reviewer_YX5H · 2025-04-02
> >
> > I thank the authors for their detailed response and I am upgrading my rating to "Accept".

---

### Official Review · Reviewer_KyTi · 2025-03-14

**Overall Recommendation:** 4

**Summary:**

This paper benchmarks the decision-making ability of several frontier multimodal models in interactive environments through in-context imitation. It investigates whether these models can be effectively prompted with few- or many-shot demonstrations to solve interactive tasks. The overall finding is that most frontier models fail to reach expert-level performance through prompting, and their performance remains independent of the number of demonstrations.

**Claims And Evidence:**

All claims are clear and supported with convincing evidence.

**Essential References Not Discussed:**

No.

**Experimental Designs Or Analyses:**

Yes.

**Methods And Evaluation Criteria:**

Yes.

**Other Comments Or Suggestions:**

No.

**Other Strengths And Weaknesses:**

While the experiments seem simple, they are well designed and presented. Even though the results are not too surprising, they are an important sanity check of the current capability of frontier models for interactive decision making.

**Questions For Authors:**

1. How do you make sense of more demonstrations having little and sometimes negative impact?
2. How should we change the training of these frontier models so that they are more effective with in-context prompting at solving interactive decision-making tasks?

**Relation To Broader Scientific Literature:**

Yes, they are related to in-context imitation of decision making with LLMs. The finding here is in line with the finding in [1] that zero-shot LMs cannot perform vision-based decision-making effectively in interactive environments.

[1] BALROG: BENCHMARKING AGENTIC LLM AND VLM REASONING ON GAMES

**Theoretical Claims:**

No.

---

> ### Author Rebuttal · Authors · 2025-04-01
>
> We thank the reviewer for their positive feedback and interesting questions. We are pleased that they think that our `claims are clear and supported with convincing evidence`, that our `experiments are well designed and presented`, and that our `results are an important sanity check of the current capability of frontier models for interactive decision making`.
>
>
> **Why do more demonstrations have little and sometimes negative impact?**
>
> At least in theory, our fully observable tasks do not require an optimal policy to attend to information more than one step in the past. Accordingly, an LM’s performance on our tasks must lie somewhere between two extremes:
> * If the LM has stored the optimal policy in its weights, then any historical information would only inform the model about the task at hand and which of its in-weights policies to select (for which even a single observation might suffice).
> * If the LM has no suitable pretrained policies in its weights and, therefore, would have to rely purely on distilling the expert behavior in the context, more demonstrations should increase performance.
>
> In practice, the LMs’ behavior lies somewhere between these extremes and, as our results show, greatly varies per model and task. Possible “mechanical” explanations for this behavior are that LMs are pretrained to attend sparsely over their context (in which case more observations would only help marginally), have a recency bias (which would make attention beyond the current/previous episode even less likely), and are usually not explicitly trained for in-context learning from examples (although it’s a somewhat emergent capability of pre-training).
>
> We mostly observe negative impact, i.e., “in-context interference” for Cheetah Run and gridworld navigation with ASCII observations. In both cases, our analysis of the percentage of illegal actions (Figures A18 and A19) suggests the problem is not due to an increased number of illegal actions but, instead, increasingly suboptimal actions. A definitive answer to what causes in-context interference would require a thorough investigation, which is beyond the scope of this paper (which introduces the benchmark), but we have added a brief discussion of these observations and open questions to the paper. Please also see the summarized findings of our investigations of failure modes (mostly appendix results) in our response to reviewer `46g3`, under “Could you include more analysis about failure modes and potential improvement strategies?”.
>
>
> **How should we change the training of these frontier models to be more effective with in-context prompting when solving interactive decision-making tasks?**
>
> While we do not have a definite answer, it seems plausible that pretraining or finetuning with data from interactive decision-making tasks, and, in particular, in-context imitation of an expert policy, would be quite effective. Another promising direction could be to focus on models capable of general in-context reinforcement learning, which is a bit different than our in-context imitation setting, but in principle, all our tasks could easily be extended by providing additional reward observations. We consider these very interesting and important directions for future research and have updated our manuscript with the above discussion.

---

### Decision · Program_Chairs · 2025-05-01

**Decision:**

Accept (poster)

**Comment:**

This paper presents a benchmark for studying the decision-making abilities of multimodal models as policies in long-context tasks, e.g., game-playing or robot control. Authors find that current models perform poorly on these tasks out-of-the-box. The paper also includes experiments to identify the utility of in-context imitation learning and different observation representations for getting models to perform better, and find some mixed results. This benchmark and its analysis would be a valuable contribution to the community. I suggest the authors to incorporate the feedback provided by the reviewers, particularly when it comes to related work and failure analysis.